# MM1.5: Methods, Analysis & Insights from Multimodal LLM Fine-tuning

**Haotian Zhang°, Mingfei Gao°, Zhe Gan°, Philipp Dufter\*, Nina Wenzel\*,**
**Forrest Huang\*, Dhruti Shah\*, Xianzhi Du\*, Bowen Zhang\*, Yanghao Li\*,**
**Sam Dodge, Keen You, Zhen Yang, Aleksei Timofeev, Mingze Xu,**
**Hong-You Chen, Jean-Philippe Fauconnier, Zhengfeng Lai, Haoxuan You,**
**Zirui Wang, Afshin Dehghan, Peter Grasch\*, Yinfei Yang†**
Apple
`{haotian.zhang2,mgao22,zhe.gan,yinfeiy}@apple.com`
°First authors; \*Core authors; †Project lead

## Abstract

We present **MM1.5**, a new family of multimodal large language models (MLLMs) designed to enhance capabilities in text-rich image understanding, visual referring and grounding, and multi-image reasoning. Building upon the MM1 architecture, MM1.5 adopts a data-centric approach to model training, systematically exploring the impact of diverse data mixtures across the entire model training lifecycle. This includes high-quality OCR data and synthetic captions for continual pre-training, as well as an optimized visual instruction-tuning data mixture for supervised fine-tuning. Our models range from 1B to 30B parameters, encompassing both dense and mixture-of-experts (MoE) variants, and demonstrate that careful data curation and training strategies can yield strong performance even at small scales (1B and 3B).[1] Through extensive empirical studies and ablations, we provide detailed insights into the training processes and decisions that inform our final designs, offering valuable guidance for future research in MLLM development.

## 1 Introduction

Multimodal Large Language Models (MLLMs) have emerged as an increasingly active research topic in recent years. Closed-source models, such as GPT-4o (Islam & Moushi, 2024), GPT-4V (OpenAI, 2024), Gemini-1.5 (Team et al., 2023; Reid et al., 2024), and Claude-3.5 (Anthropic, 2023), have demonstrated remarkable capabilities in advanced multimodal understanding. Meanwhile, open-source models, such as the LLaVA series of work (Liu et al., 2023b;a; 2024a; Li et al., 2024c), InternVL2 (Chen et al., 2024b), Cambrian-1 (Tong et al., 2024a) and Qwen2-VL (Bai et al., 2023; team, 2024), are rapidly narrowing the performance gap. There has also been growing interest in developing models capable of understanding single-image, multi-image, and video data using a single set of model weights (Li et al., 2024c). Further discussion of recent works is in Appendix A.1.

Building upon MM1 (McKinzie et al., 2024), we introduce MM1.5, a new family of MLLMs carefully designed to enhance a set of core capabilities. Specifically, we focus on the following aspects.

- **OCR.** MM1.5 supports arbitrary image aspect ratios and resolutions of up to 4 Megapixels. By incorporating carefully selected OCR data to enhance text comprehension across different training stages, MM1.5 excels at understanding text-rich images.

- **Visual referring and grounding.** MM1.5 offers robust, fine-grained image understanding, extending beyond text prompts to interpret *visual* prompts such as points and bounding boxes. Moreover, MM1.5 can generate grounded responses by grounding text output with image bounding boxes. This capability is notably under-explored in most open-source models (*e.g.*, LLaVA-OneVision (Li et al., 2024c) and Phi-3-Vision (Abdin et al., 2024b)), and even in strong proprietary models like GPT-4o, which rely on set-of-mark (SoM) prompting (Yang et al., 2023) to reference image regions.

---

[1]Additionally, we introduce two specialized variants: MM1.5-Video, designed for video understanding, and MM1.5-UI, tailored for mobile UI understanding, detailed in Appendix A.7 and A.8 correspondingly.

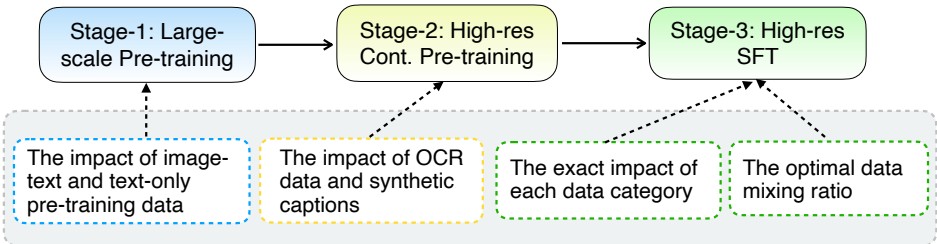

Figure 1: Recipe for building MM1.5. Model training contains three stages: ($i$) large-scale pre-training with low-resolution images (378×378), ($ii$) continual pre-training with high-resolution (up to 4 Megapixels) OCR data and synthetic captions, and ($iii$) supervised fine-tuning (SFT). At each stage, we aim to identify the optimal data mix and assess the impact of each data type.

- **Multi-image reasoning.** MM1.5 benefits from large-scale interleaved pre-training, resulting in strong in-context learning and multi-image reasoning capabilities right out of the box. We further improve its capabilities via supervised fine-tuning (SFT) on additional multi-image data.

Our primary focus is on the most efficient model scales, 1B and 3B, together with their corresponding Mixture-of-Experts (MoE) variants, and demonstrates that even relatively small MLLMs can achieve competitive performance on various downstream tasks. Furthermore, we demonstrate that the MM1.5 recipe exhibits strong scaling behavior all the way to 30B parameters, achieving competitive performance across a wide range of benchmarks.

Building performant MLLMs is a highly empirical endeavor. While the overarching goal and the high-level training procedure are well-defined, the finer details of their execution remain unclear. In developing MM1.5, we choose to retain the same model architecture as MM1 (McKinzie et al., 2024), enabling us to focus on refining and investigating the intricacies of our data-centric training recipes. Our attention is centered on the following key aspects:

- **Continual Pre-training.** We introduce an additional high-resolution continual pre-training stage preceding the SFT stage, which we found crucial for boosting text-rich image understanding performance. We ablate the impact of two kinds of data: ($i$) text-rich OCR data, focusing on detailed transcription of text within images; and ($ii$) high-quality synthetic image captions.

- **SFT.** While considerable prior work discusses SFT data for MLLMs, there is still limited exploration into how each category of SFT data can affect the final model's performance. In particular, the impact of data supporting each capability on other capabilities is understudied. We conduct extensive ablations to identify trade-offs and synergies, ultimately constructing a mixture from public datasets that contributes to well-balanced performance across a wide set of capabilities.

- **Dynamic High-resolution.** Furthermore, for *high-resolution* image encoding, we follow the popular any-resolution approach, dynamically dividing the image into sub-images (Li et al., 2023i; Zhang et al., 2024b), and conduct thorough ablations to refine key details in our design.

Unlike most open-source models focusing solely on SFT (Liu et al., 2023b;a; 2024a), MM1 demonstrated strong zero-shot and few-shot learning capabilities through large-scale pre-training. In developing MM1.5, we aim to retain these strengths and more effectively transfer them to the SFT stage. To achieve this, We further extend MM1's pre-training by exploring the impact of text-only data and optimizing the ratio of different pre-training data types. This approach improves performance on knowledge-intensive benchmarks and enhances overall multimodal understanding capabilities.

Our main contributions are summarized as follows: ($i$) We introduce MM1.5, a family of MLLMs that include both dense models (ranging from 1B to 30B) and MoE variants. ($ii$) We conduct a thorough empirical study detailing the process and decisions leading to our final design choices. ($iii$) MM1.5 excels in handling a wide range of multimodal tasks, from general-domain to text-rich image understanding, coarse- to fine-grained understanding, and single- to multi-image reasoning.

## 2 RECIPE FOR BUILDING MM1.5

In this work, beyond pre-training and SFT stages as in MM1 (McKinzie et al., 2024), we introduce a continual pre-training stage with high-quality OCR data and synthetic captions. As outlined in Figure 1, to obtain the best data recipe,

- We first present comprehensive ablations of our SFT data mixture (Section 2.2). We categorize the SFT data into multiple groups based on the capabilities they aim to support. We carefully evaluate the impact of datasets from each category and adjust the ratio to balance different core capabilities.

- To further enhance model performance, especially for text-rich image understanding, we further ablate the data choices for continual pre-training (Section 2.3). This includes 45M rich OCR data and 7M high-quality captions from public or generated by a MM1-based image captioner.

- We also provide detailed ablation regarding dynamic image splitting, also known as AnyRes (Liu et al., 2024a) (Section 2.4), for high-resolution image comprehension.

Finally, to enhance performance on knowledge-heavy benchmarks like MMMU (Yue et al., 2023a), we further study the impact of pre-training data (Appendix A.2).

## 2.1 Empirical Setup for Ablations

Unless otherwise noted, we follow the default settings below in our ablation studies.

**Model architecture and data preprocessing.** We follow MM1 (McKinzie et al., 2024) and use the same architecture, focusing on the 3B dense model for all the ablation studies in this section. Specifically,

- Static image splitting (Lin et al., 2023b) is enabled with 4 sub-image splits (plus an overview image), and each sub-image is resized to 672×672 resolution via position embedding interpolation. Note that we did not use dynamic image splitting during ablation for faster iteration of experiments.

- As to the encoding of multi-image data, we enable image splitting only when the current training sample contains fewer than three images to avoid excessively long sequence lengths.

- Similar to capabilities introduced in Ferret (You et al., 2023), MM1.5 directly supports referring and grounding. When requested, MM1.5 can produce bounding boxes in its textual output to ground its responses. Additionally, the model can interpret references to points and regions in the input image in the form of referring coordinates and bounding boxes.

- The CLIP image encoder and the LLM backbone are based on in-house models, with C-Abstractor (Cha et al., 2024) serving as the vision-language connector.

**Model optimization.** For both continual pre-training and SFT, we set the batch size as 256. We use the AdaFactor optimizer with a peak learning rate of 1e-5 and a cosine decay of 0. For continual pre-training, we train a maximum of 30k steps. During SFT, all models are optimized for one epoch.

**Continual pre-training.** Models are initialized with the MM1 pre-trained checkpoint. By default, we conduct continual pre-training on 45M high-resolution OCR data (including PDFA, IDL, Rendered-text (Laurençon et al., 2024a) and DocStruct-4M (Hu et al., 2024a)) at this stage. In each training batch, data is equally sampled from those four datasets. Similar to the SFT stage, we use static image splitting, dividing each image into five sub-images, with each sub-image resized to 672×672 resolution. We find that this high-resolution setup is essential for continual pre-training.

**SFT data categorization.** Grouping datasets into categories can be helpful for data balancing and simplifying the analysis (Laurençon et al., 2024a; Tong et al., 2024a). At a high level, we cluster datasets into *single-image*, *multi-image*, and *text-only* categories based on the number of images presented in each example. For the single-image group, we further classify each dataset into the following sub-categories: *general*, *text-rich*, *refer&ground*, *science*, *math* and *code*. See Table 5 in Appendix A.3 for the details of each category used for the ablation study, and Figure 8 for an overview of the group categories.

**Evaluation benchmarks.** We group our benchmarks into categories based on what capabilities a benchmark primarily measures. Our benchmark groups include general, text-rich, refer&ground, knowledge, and multi-image. See Table 6 in Appendix A.4 for more details. We propose *Category Average Score*, the average score of all benchmark numbers for each sub-category, to represent the average performance on that capability. We focus on the categories of general, text-rich, and knowledge, as these capabilities are widely considered essential for MLLMs. To evaluate a model's impact on these capabilities, we refer to a *MMBase* score, defined as the average scores on general, text-rich, and knowledge categories. Details of the evaluation metrics are provided in Appendix A.4.

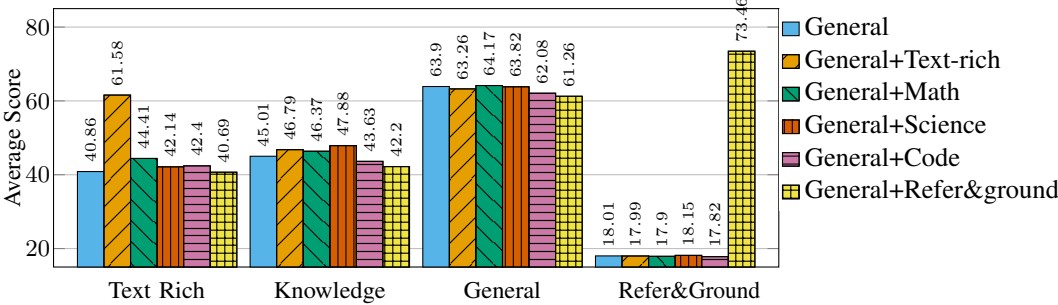

Figure 2: Impact of different SFT data categories to different model capabilities (general, text-rich, knowledge, and refer&ground). Text-rich data significantly improves text-rich and knowledge benchmarks on average. Science data improves knowledge average score. Referring and grounding data enables this capability.

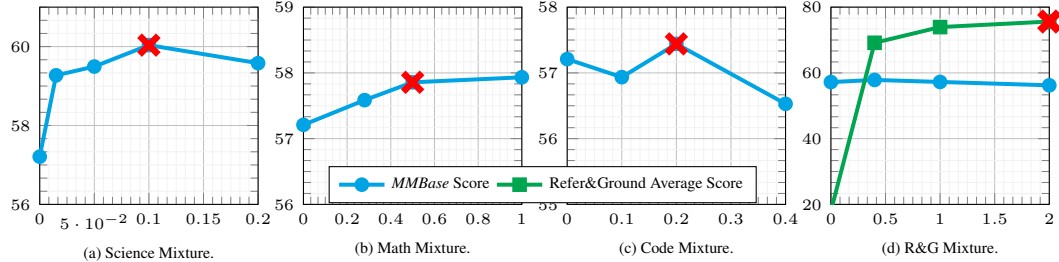

Figure 3: Impact of $\alpha$ ($x$-axis) for different data categories to a model's different capabilities. The selected ratio is marked with red "x". $\alpha$ denotes the data ratio of the target category (science, math, code, refer&ground (R&G)) when compared with the general category.

## 2.2 SFT ABLATIONS

To determine the optimal SFT recipe, we first study the impact of different data categories in Section 2.2.1, followed by investigating how to best mix all the data in Section 2.2.2.

### 2.2.1 IMPACT OF DIFFERENT DATA CATEGORIES

In this subsection, we focus on evaluating the single-image data category. We begin by assessing the general data category and then progressively evaluate the impact of adding other sub-categories individually. During training, we mix data from different sub-categories and construct each training batch by randomly sampling data from the corresponding mixture. We compare models using each capability with the *Category Average Score*.

Our results are summarized in Figure 2. We observe that adding text-rich data can significantly improve the performance on text-rich and knowledge benchmarks. The inclusion of math data follows a similar trend, though we observe a lesser degree of improvement in the text-rich average score. When science data is added, we observe the expected improvement in the knowledge benchmarks, alongside a minor improvement in text-rich performance. Adding the code category yields a slight increase in the text-rich average score, while the performance on other benchmarks does not improve. Including the refer&ground data instills the model with referring and grounding capability, but we also observe slight regression in all other capability categories.

### 2.2.2 DATA MIXTURE RATIO STUDY

We first study the mixing ratio within the single-image categories. Since directly mixing the general and text-rich data based on their data sizes shows strong results across a variety of benchmarks (see Figure 2), we use this combination as the starting point to study how to mix other categories to this set. Then, we combine the entire single-image set with multi-image and text-only sets with sampling weights of $w_{\text{single}}$, $w_{\text{multi}}$ and $w_{\text{text}}$, respectively, where $w_{\text{single}} + w_{\text{multi}} + w_{\text{text}} = 1$.

**Mixture of single-image data.** Directly mixing all datasets from different categories may not be ideal due to imbalanced numbers of data samples across different sub-categories. For example, the

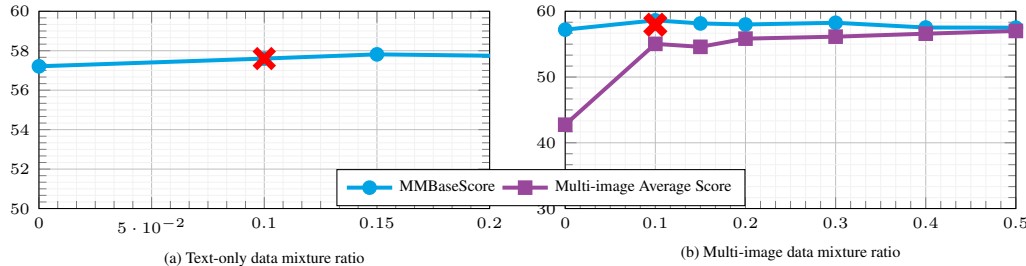

(a) Text-only data mixture ratio       (b) Multi-image data mixture ratio

Figure 4: Impact of the mixing ratio for text-only and multi-image SFT data. The selected ratio is marked with red "x".

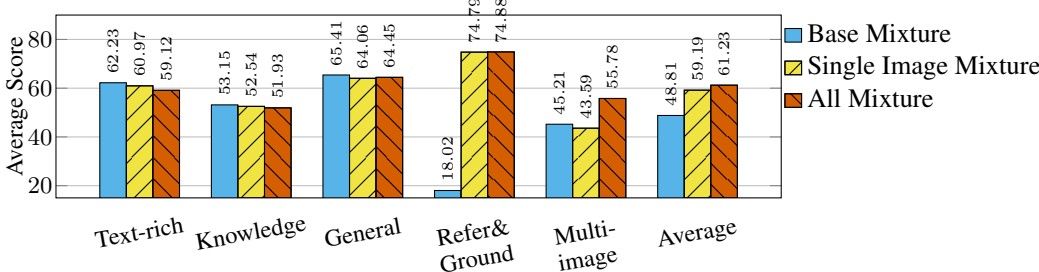

Figure 5: Ablation study of mixing all the SFT data. *Base Mixture* denotes general, text-rich and knowledge (science, math and code). The "Average" column represents the performance averaged across the preceding five benchmark categories.

size of the general data category is around $68\times$ the size of the science data category. In this study, we use the general data category as the reference, and upsample/downsample data from a target category, such that in each training batch, the data ratio from the general and target category is $1{:}\alpha$.

To measure the average impact of $\alpha$, we propose *MMBase* score, an average over general, text-rich, and knowledge average scores, for model comparison. As shown in Figure 3, we vary the $\alpha$ for different data categories. For science, math, and code categories, we find the best ratio of $\alpha$ to be 0.1, 0.5, and 0.2, respectively. As shown in Section 2.2.1, the refer&ground data is the main driver for improving referring and grounding benchmarks. Therefore, besides the *MMBase* score, we also include the Refer&Ground average score as another metric for the $\alpha$ selection. As summarized in Figure 3(d), the *MMBase* score will drop slightly, while the Refer&Ground average score increases significantly. With that, we select $\alpha = 2.0$ as a good trade-off.

**Mixture of single-image, multi-image, and text-only data.** Now, we study the mixture ratios, $w_{single}$, $w_{multi}$ and $w_{text}$. Enumerating all combinations between the three ratios will incur significant computational cost. Therefore, we instead separately ablate $w_{text}$ and $w_{multi}$ for text-only and multi-image data, respectively, to evaluate how sensitive our model is to these ratios. Finally, $w_{single}$ is determined by $1 - w_{text} - w_{multi}$.

Similar to the single-image mixture study, we also start with the combination of general and text-rich data and enumerate different values for $w_{multi}$ and $w_{text}$. For text-only data, we tested $w_{text}$ from 0 to 0.2. Figure 4(left) shows that varying different values for $w_{text}$ has minor effects on the model's base capabilities in general. We select $w_{text} = 0.1$ to allocate a higher weight for single-image data for potential performance improvements.

For multi-image data, we use the multi-image average score (evaluated on multi-image benchmarks in Table 6) as an additional metric to assess a model's capability of handling multi-image tasks. Results are summarized in Figure 4(right). We observe that increasing the sampling ratio of multi-image data would introduce a performance drop of the base capabilities as indicated by the decreased number of the *MMBase* score, while the multi-image average score increases. We select $w_{multi} = 0.1$ since it introduces a surge in the multi-image average score.

**Mixing multiple categories.** Based on the studies above, we present three mixtures, the *Base* mixture, *Single-image* mixture, and *All* mixture, and analyze their trade-offs. The *Base* mixture includes the

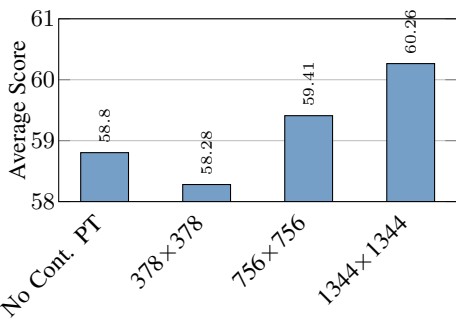 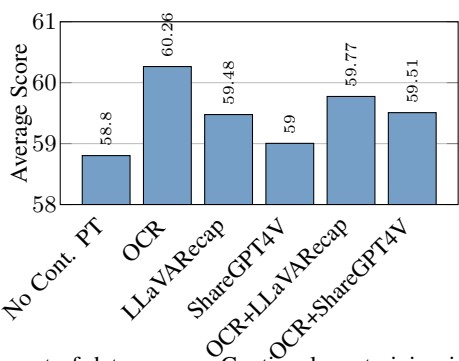

(a) Impact of input resolution. OCR data is used for all the continual pre-training experiments.

(b) Impact of data source. Continual pre-training is conducted in the high-resolution (1344×1344) setting.

Figure 6: Ablation study of continual pre-training. Average Score indicates the *MMBase* score. Cont. PT denotes continual pre-training.

general, text-rich, science ($\alpha_{\text{science}} = 0.1$), math ($\alpha_{\text{math}} = 0.5$) and code ($\alpha_{\text{code}} = 0.2$) data groups. The *Single-image* mixture additionally adds refer&ground data ($\alpha_{\text{rg}} = 2.0$) to the *Base* mixture. *All* mixture includes all data from single-image, multi-image, and text-only data, with $w_{\text{single}} = 0.8$, $w_{\text{multi}} = 0.1$, and $w_{\text{text}} = 0.1$.

Our results are summarized in Figure 5. The first three columns indicate that including refer&ground and multi-image data slightly reduces average performance on text-rich, knowledge, and general benchmarks. The fourth column shows that adding refer&ground data significantly boosts referring and grounding performance, while the fifth column highlights that adding multi-image data greatly improves multi-image benchmarks. The final column reveals that our optimized mixture achieves the best overall performance, balancing all capabilities across benchmarks.

## 2.3 CONTINUAL PRE-TRAINING ABLATIONS

Unless otherwise specified, we use OCR data (45M in total), including PDFA, IDL, Rendered-text (Laurençon et al., 2024a) and DocStruct-4M (Hu et al., 2024a) in a high-resolution setting (1344×1344) for continual pre-training. During the SFT stage, all continual pre-trained models in this section are fine-tuned with data from the *Base Mixture* including general, text-rich, knowledge (science, math, and code) with their selected mixture ratios as described in Section 2.2.2.

**Impact of image resolution.** Intuitively, higher-resolution images are preferable when training with OCR data. We first ablate the impact of image resolution during this stage by setting up two baselines, continual pre-training with 378×378 and 756×756 resolutions, respectively. For the former, we disabled both image splitting and position embedding interpolation (our CLIP image encoder natively supports image resolution of 378×378). For the latter, we enabled image splitting and turn-off position embedding interpolation. The results are shown in Figure 6(a). Note that the final SFT stage always uses image resolution 1344×1344 across these experiments, so the training only differs with respect to the image resolution used in continual pre-training.

We can clearly see that using a setting of 1344×1344 image resolution for continual pre-training achieves the best overall performance. Decreasing resolution consistently leads to lower final scores. In particular, continual pre-training with 378×378 resolution can underperform a model without continual pre-training. We hypothesize this is due to insufficient visible detail at lower resolutions, which may hinder the model's ability to effectively learn from the document-based OCR data in the continual pre-training mixture.

**Impact of OCR data and synthetic captions.** Besides OCR data, high-quality synthetic image captions (Chen et al., 2023c; Li et al., 2024a) are also widely considered useful for pre-training. To study its impact, we use our default setting except for the data used in continual pre-training. We study two synthetic caption datasets: LLaVA-Recap-3M (Li et al., 2024a) and ShareGPT4V-PT (Chen et al., 2023c), and their combination with our OCR data. When we combine ShareGPT4V-PT or LLaVA-Recap-3M with our OCR data, we equally sample data from individual datasets in each training batch. Results are presented in Figure 6(b). We observe that all continual pre-trained models

Table 1: Ablation on the image resolution and the number of image tokens used in dynamic image splitting. $n$ denotes the total number of sub-images. Row 3: $(n_{\min}, n_{\max}) = (4, 4)$; Row 4-7: $(n_{\min}, n_{\max}) = (9, 9)$. Image encoder resolution: $(i)$ $378 \times 378$: no position embedding interpolation; $(ii)$ $672 \times 672$: with position embedding interpolation.

| Row # | Mode | $n$ | #image tokens (per sub-img / total) | Image Enc. Resolution | Effective Resolution | Text-rich | Knowledge | General | Refer & Ground | Average |
|---|---|---|---|---|---|---|---|---|---|---|
| 1 | Static | 1 | 144/144 | 672×672 | 0.45MP | 49.4 | 53.6 | 62.6 | 71.3 | 59.2 |
| 2 |  | 5 | 144/720 | 672×672 | 1.8MP | 57.7 | 53.8 | 64.4 | 74.8 | 62.7 |
| 3 |  | 5 | 144/720 | 672×672 | 1.8MP | 58.6 | 53.7 | 64.1 | 74.0 | 62.5 |
| 4 |  | 10 | 81/810 | 378×378 | 1.3MP | 57.6 | 53.3 | 62.9 | 74.0 | 62.0 |
| 5 | Dynamic | 10 | 81/810 | 672×672 | 4.1MP | 58.3 | 53.8 | 64.3 | 74.9 | 62.8 |
| 6 |  | 10 | 144/1440 | 378×378 | 1.3MP | 58.5 | 54.0 | 63.2 | 74.5 | 62.6 |
| 7 |  | 10 | 144/1440 | 672×672 | 4.1MP | 59.8 | 54.0 | 64.5 | 75.2 | 63.3 |

perform better than the baseline without continual pre-training. However, we did not find conclusive evidence that these high-quality synthetic captions improved performance over the arguably simpler OCR data. While prior studies (Li et al., 2024c) show synthetic captions boost performance, our results indicate further investigation into their exact impact is needed.

Therefore, we further investigate the impact of synthetic captions generated through self-training for even larger scales (up to 7M) and more controllable styles, using a pre-trained MM1 model fine-tuned on human-annotated captions, similar to (Fang et al., 2024). This new dataset showed some promise in certain settings, see Appendix A.5 for details. We defer further study of this topic to future work.

## 2.4 DYNAMIC IMAGE SPLITTING ABLATIONS

**Dynamic image splitting.** Processing high-resolution images is essential for text-rich image understanding. In *static* image splitting (Lin et al., 2023b), images are split into a rigid $2\times2$ grid, which is often inefficient. Low-resolution images are splitted without any need, and images with non-square aspect ratios can result in sub-images being padding only. Therefore, we adopt a dynamic image splitting approach, which is common in the literature (Li et al., 2024a; Dong et al., 2024b; Hu et al., 2024a; Lin et al., 2023b; Xu et al., 2024c; Zhang et al., 2024d), for MM1.5.

Given a minimum and maximum number of sub-images, $n_{\min}$ and $n_{\max}$, consider the set of all candidate grids $G = \{(n_h, n_w) \in \mathbb{N} \,|\, n_{\min} \leq n_h \cdot n_w \leq n_{\max}\}$. Further, consider the resolution of the vision encoder $r$, and an input image resolution $(h, w)$. If there is a grid that can cover the image, we choose the grid that minimizes the amount of padding after longer side resizing to the grid, *i.e.*, $g^* = \arg\min_{(n_h, n_w)=g\in G} n_h n_w r^2 - h_g w_g$, subject to $n_h r \geq h_g \geq h$ and $n_w r \geq w_g \geq w$, where $h_g, w_g$ denote the image height and width after longer side resizing the candidate grid. If no such grid exists, we choose the one that minimizes the resolution loss due to scaling the image down and fully covers the longer side resized image. Assume we allow up to 4 sub-images. With a static image splitting approach, all images use the grid $(2, 2)$. The dynamic splitting approach instead allows for the following grids: $\{(1, 1), (1, 2), (2, 1), (1, 3), (3, 1), (1, 4), (4, 1), (2, 2)\}$.

**Global-Local Format.** In addition to the sub-images, we always feed the original image with a longer side resized to the encoder resolution $r$ to the model. This ensures that the model has a global understanding of the image. If the grid is $(1, 1)$, we omit the overview image. We consider two variants: $(i)$ **before**: the overview image is put before the sub-images; $(ii)$ **after**: the overview image is put after the sub-images. These variants yield different results because an autoregressive mask is used in the LLM decoder, and as such, the choice determines whether the decoder can attend to the overview image when processing the sub-images $(i)$ or attend to the sub-images when processing the overview image $(ii)$.

**Sub-image position indicator.** Given that an input image is dynamically split into multiple sub-images, we explore whether it is helpful to indicate the position of each sub-image in the original high-resolution image to ensure the model can understand the original 2D image structure. Specifically, we consider two methods.

- **Index.** A tuple of $(k, i, j)$ is used to represent sub-image position information, where $k$ is the zero-indexed image number in the example (assuming there can be multiple images in a training sample), $i$ and $j$ are the one-index row and column id, *e.g.*, $(0, 0, 0)$ is the overview image of image 0, and $(0, 2, 1)$ is the sub-image in the second row and first column, for image 0.

Table 2: Ablation on the image grid configuration $(n_{\min}, n_{\max})$ used in dynamic image splitting.

| Row # | $(n_{\min}, n_{\max})$ Train | Inference | DocVQA | InfoVQA | Text-rich | Knowledge | General | Refer & Ground | Average |
|---|---|---|---|---|---|---|---|---|---|
| *3B Model Comparison* | | | | | | | | | |
| 1 | $(4, 4)$ | $(4, 4)$ | 73.2 | 48.3 | 58.6 | 53.3 | 64.1 | 74.0 | 62.5 |
| 2 | $(4, 9)$ | $(4, 9)$ | 75.7 | 53.8 | 60.0 | 54.0 | 63.9 | 74.6 | 63.1 |
| 3 | $(4, 16)$ | $(4, 16)$ | 76.3 | 55.2 | 60.7 | 53.4 | 64.0 | 73.8 | 63.0 |
| 4 | $(1, 9)$ | $(1, 9)$ | 76.2 | 54.1 | 60.4 | 53.7 | 62.5 | 71.5 | 62.0 |
| 5 | $(4, 4)$ | $(4, 9)$ | 73.4 | 52.9 | 59.7 | 53.5 | 63.8 | 74.0 | 62.8 |
| 6 | $(4, 4)$ | $(4, 16)$ | 72.3 | 53.5 | 59.6 | 53.8 | 63.5 | 74.0 | 62.7 |
| 7 | $(4, 4)$ | $(1, 9)$ | 73.5 | 52.7 | 59.8 | 50.7 | 62.6 | 24.5 | 49.4 |
| *7B Model Comparison* | | | | | | | | | |
| 8 | $(4, 4)$ | $(4, 4)$ | 77.0 | 54.3 | 64.5 | 61.1 | 66.8 | 77.7 | 67.5 |
| 9 | $(4, 9)$ | $(4, 9)$ | 81.7 | 62.1 | 67.4 | 60.1 | 66.6 | 78.0 | 68.0 |
| 10 | $(4, 16)$ | $(4, 16)$ | 83.3 | 64.1 | 68.0 | 58.7 | 67.7 | 77.2 | 67.9 |

Table 3: Ablation on the sub-image position indicator and the position of the overview image. We set $(n_{\min}, n_{\max}) = (4, 4)$ for experiments.

| Row # | Sub-img pos. indicator | Overview image pos. | DocVQA | InfoVQA | Text-rich | Knowledge | General | Refer & Ground | Average |
|---|---|---|---|---|---|---|---|---|---|
| 1 | none | before | 73.2 | 48.3 | 58.6 | 53.5 | 64.1 | 74.0 | 62.5 |
| 2 | seps | before | 74.3 | 49.7 | 58.8 | 53.0 | 63.8 | 74.5 | 62.5 |
| 3 | index | before | 73.4 | 48.6 | 58.6 | 52.7 | 63.4 | 74.8 | 62.4 |
| 4 | none | after | 73.3 | 49.7 | 59.2 | 54.3 | 64.1 | 73.8 | 62.8 |

- **Seps.** Instead of using indexes, we use three text tokens. Specifically, ':' is the overview image indicator, ',' is the column separator, and '<n>' is the row separator. The latter two tokens are inserted between the set of image tokens corresponding to each sub-image so that the original 2D image structure can be recovered from the flattened image token sequence.

**Inference for higher resolution.** The tuple $(n_{\min}, n_{\max})$ is used to decide the dynamic image splitting configuration for model training. During inference, it is possible to support even higher-resolution image processing simply by increasing these parameters. For example, we explore training at $(n_{\min}, n_{\max}) = (4, 9)$ to save model training compute, while during inference, we use $(n'_{\min}, n'_{\max}) = (4, 16)$ to process images at even higher effective resolutions.

**Findings.** We use the final *Single-image Mixture* as our default experiment setting, including general, text-rich, knowledge (science, math and code), and refer&ground data. For fast iteration of experiments, all the models are initialized with the MM1 pre-trained checkpoint without continual pre-training. Following Figure 2, we report the average performance on text-rich, knowledge, general, and refer&ground benchmarks. Our findings are summarized as follows.

- Dynamic image splitting outperforms static splitting in text-rich tasks. Both image resolution and the number of sub-images, along with image token counts, are important (Table 1).

- Regarding image grid configuration (Table 2), dynamic image splitting with a larger $n_{\max}$ significantly improves performance on tasks with unusual aspect ratios, such as DocVQA and InfoVQA, especially when trained natively for it. Grounding performance is highly sensitive to grid size changes, and larger LLM backbones yield greater performance gains, with the 7B model showing larger improvements than the 3B model.

- In terms of position indicators (Table 3), they are not strictly necessary for text-rich tasks, though they can be beneficial for specific tasks like DocVQA and InfoVQA. Additionally, placing the overview image after the sub-images improves performance, as it allows the decoder attention mask to attend to all sub-images more effectively.

- In practice, dynamic image splitting does not significantly increase the number of sub-images to process. In a sample of 100k examples, static splitting generates 500k sub-images, while dynamic splitting with $(n_{\min}, n_{\max}) = (4, 9)$ produces barely more, only 539k images in total.

## 3 FINAL MODEL AND TRAINING RECIPE

We collect the results from the previous ablations to determine the final recipe for MM1.5 training:

- **Architecture.** We use the same model architecture as MM1 (McKinzie et al., 2024).

- **Data and training pipeline.** As summarized in Figure 1, MM1.5 is trained in three stages:
  - **Pre-training.** The pre-training data comprises three parts: ($i$) 2B image-text pairs, ($ii$) 600M interleaved image-text documents with 1B images in total, and ($iii$) text-only data with 2T tokens. Except for the updated text-only data, the data remains unchanged from MM1 (McKinzie et al., 2024). However, the data ratio has been adjusted from 45:45:10 to 50:10:40, significantly downweighting the interleaved data (from 45% to 10%) while increasing the proportion of text-only data (from 10% to 40%) as discussed in Section A.2.
  - **Continual Pre-training.** We use 45M OCR data to enhance text-rich image understanding. Notably, we do not include additional synthetic image captions based on empirical results.
  - **SFT.** We use the data illustrated in Figure 8 and adopt the mixing ratios studied in Section 2.2.2. Our final mixture consists of 80% single-image data, 10% multi-image data, and 10% text-only SFT data. The single-image data can be further categorized into 37.2% text-rich data, 22.5% refer&ground data (VQA data enriched with bounding boxes and/or point coordinates), 11.3% general data, 5.6% math data, 2.3% code data, and 1.1% science data, totaling 80% of all data.

- **Dynamic high-resolution.** We set the image grid configuration $(n_{\min}, n_{\max}) = (4, 9)$, using an index for the sub-image position indicator and placing the overview image after the sub-images. Dynamic image splitting is only enabled when the current training sample has fewer than three images. The supported resolution reaches up to 4 Megapixels (approximately 2016×2016 for a square image, or 6048×672 for a long image).

We use the same image encoder and the LLM backbone from MM1 (McKinzie et al., 2024), and keep them *unfrozen* during all the model training stages. For pre-training, we follow the exact same learning rate schedule as in MM1 (McKinzie et al., 2024) and 200k training steps with sequence length 4096. For continual pre-training, we use a peak learning rate of 1e-5 with the cosine decay and 30k training steps for all the models (from 1B to 30B). For SFT, we use a peak learning rate of 2e-5 and 23k training steps for all the models.[2]

**Mixture-of-Experts (MoE).** We introduce two MoE models, a 1B-MoE and a 3B-MoE, with 64 experts replacing dense layers every two layers. We use the same hyperparameters as those applied to the dense models for both scales and top-2 gating with a 0.01 load balance loss to encourage a better expert load balance and a 0.001 router z-loss for training stability (Lepikhin et al., 2021).

### 3.1 RESULTS

We evaluate our MM1.5 models across 35 multimodal benchmarks using an internal fork of lm-eval-harness (Gao et al., 2023), covering task categories ranging from general multimodal understanding, knowledge, text-rich, referring and grounding, multi-image reasoning, to in-context learning. Detailed results for each capability at various model sizes are further summarized in Table 7, 8, 9, 10, and 11, respectively. Below, we highlight a few key observations from Table 4.

**MM1.5 represents a major upgrade over MM1.** It delivers improvements across all model sizes and nearly all benchmarks, often by a substantial margin. For instance, MM1.5-30B boosts the MathVista from 39.4 to 55.6, DocVQA from 75.8 to 91.4, and InfoVQA from 47.3 to 67.3. It also offers much enhanced multi-image reasoning capability and introduces new capabilities not present in MM1, such as visual referring & grounding.

**Both Dense and MoE model scaling are effective.** First, scaling the dense model from 1B to 30B consistently improves performance, with benchmarks like AI2D increasing from 59.3 to 77.2 (see Table 7). Second, both the 1B and 3B MoE models outperform their dense counterparts. Notably, the MM1.5-3B-MoE model can even surpass the MM1.5-7B model in knowledge, general, visual referring and grounding, and multi-image benchmarks, though it falls slightly behind on text-rich benchmarks. This suggests that MoE models show strong potential in integrating diverse capabilities compared to dense models.

---

[2]All models are trained using the AXLearn framework `https://github.com/apple/axlearn`

Table 4: Comparison with SOTA models on the selected benchmarks. Comparison of more benchmarks and broader baselines can be found in Table 7,8,9, 10 and 11in the Appendix.

| Model | Text-rich | | | Knowledge | | General | | Refer&Ground | | Multi-image | In-context |
|---|---|---|---|---|---|---|---|---|---|---|---|
| | TextVQA (val) | DocVQA (test) | InfoVQA (test) | MMMU (val) | MathV (testmini) | SEED[I] | RealWorldQA | RefCOCO avg | LVIS avg | NLVR2 (val) | VL-ICL avg |
| *1B Model Comparison* | | | | | | | | | | | |
| LLaVAOneVision-0.5B | – | 70.0 | 41.8 | 31.4 | 34.8 | 65.5 | 55.6 | – | – | 63.4 | – |
| SPHINX-Tiny | 57.8 | 53.0 | 26.3 | – | 26.4 | – | – | 77.2 | – | – | – |
| MM1-1B | 68.2 | 68.4 | 38.5 | 33.2 | 31.1 | 65.6 | 51.2 | – | 51.5 | 50.9 | 34.3 |
| MM1.5-1B | 72.5 | 81.0 | 50.5 | 35.8 | 37.2 | 70.2 | 53.3 | 81.4 | 62.2 | 79.0 | 51.0 |
| MM1.5-1B-MoE | 76.1 | 84.8 | 55.9 | 41.2 | 42.9 | 71.4 | 57.8 | 83.9 | 64.1 | 83.2 | 56.0 |
| *3B Model Comparison* | | | | | | | | | | | |
| IntenVL2-2B | 73.4 | 86.9 | 58.9 | 36.3 | 46.0 | 70.9 | 57.4 | 77.7 | 51.1 | 67.4 | 18.5 |
| MiniCPM-V 2.0-3B | 74.1 | 71.9 | 37.6 | 38.2 | 38.7 | 67.1 | 55.8 | – | 48.0 | – | – |
| Phi-3-Vision-4B | 70.1 | 83.3 | 49.0 | 40.4 | 44.5 | 71.8 | 59.4 | 38.1 | 54.2 | 53.6 | 19.5 |
| MM1-3B | 71.9 | 75.2 | 44.7 | 33.9 | 32.0 | 68.8 | 55.8 | – | 53.4 | 51.7 | 37.0 |
| MM1.5-3B | 76.5 | 87.7 | 58.5 | 37.1 | 44.4 | 72.4 | 56.9 | 85.6 | 67.9 | 83.8 | 56.3 |
| MM1.5-3B-MoE | 76.8 | 85.0 | 53.6 | 42.9 | 46.9 | 73.3 | 60.7 | 86.2 | 66.9 | 86.0 | 59.6 |
| *7B Model Comparison* | | | | | | | | | | | |
| MM1-7B | 72.8 | 76.8 | 45.5 | 37.0 | 35.9 | 69.9 | 55.7 | – | 53.2 | 59.9 | 52.8 |
| MM1.5-7B | 76.5 | 88.1 | 59.5 | 41.8 | 47.6 | 73.4 | 62.5 | 86.6 | 66.4 | 86.9 | 56.0 |
| *30B Model Comparison* | | | | | | | | | | | |
| MM1-30B | 73.5 | 75.8 | 47.3 | 44.7 | 39.4 | 72.1 | 59.4 | – | 53.1 | 63.1 | 52.1 |
| MM1.5-30B | 79.2 | 91.4 | 67.3 | 47.4 | 55.6 | 75.0 | 69.0 | 90.1 | 73.2 | 90.6 | 77.6 |

**MM1.5-1B is the state-of-the-art model at the 1B scale.** While few models are available at this scale, MM1.5-1B clearly outperforms comparable models such as SPHINX-Tiny (Gao et al., 2024). For reference, MM1.5-1B also significantly surpasses LLaVAOneVision-0.5B (Li et al., 2024c) (*e.g.*, ScienceQA: 67.2 vs. 82.1, DocVQA: 70.0 vs. 81.0, see Table 7 and 8), but it should be stressed that this is of course an even smaller model and as such cannot be directly compared.

**MM1.5-3B outperforms MiniCPM-V 2.0 and is competitive with InternVL2 and Phi-3-Vision.** Compared to MiniCPM-V 2.0, MM1.5-3B achieves superior results on MathVista (38.7 vs. 44.4) and DocVQA (71.9 vs. 87.7) while also supporting visual referring and grounding, which MiniCPM-V lacks. It surpasses InternVL2-2B on general VQA and, although Phi-3-Vision (4.2B) has an edge on knowledge-based tasks like AI2D as in Table 7, MM1.5-3B excels on text-rich benchmarks (e.g., DocVQA: 83.3 vs. 87.7) and outperforms Phi-3-Vision in referring and grounding (see Table 9), as well as in-context learning tasks (see Table 10).

**MM1.5 excels in visual referring and grounding.** While most SOTA models focus on improving performance across general, knowledge, and text-rich benchmarks, few have integrated fine-grained image grounding and referring ability into their design. Even GPT-4o relies on set-of-mark prompting to demonstrate visual grounding capabilities. As shown in Table 9, MM1.5-3B outperforms Ferret-7B and is on par with Ferret-13B, both of which are fine-tuned specifically for referring and grounding tasks. Notably, our model inherently possesses these capabilities while still excelling in other areas.

**MM1.5 excels in multi-image reasoning and in-context learning.** As shown in Table 11, the MM1.5-1B model outperforms LLaVAOneVision-0.5B at the 1B scale. Similarly, at the 3B scale, MM1.5-3B significantly surpasses Phi-3-Vision. Additionally, we evaluate MM1.5's zero-shot transfer capability for video understanding using MVBench (Li et al., 2024h), a benchmark designed for video tasks. Moreover, we evaluate MM1.5's ability of multimodal in-context learning on VL-ICL benchmark (Zong et al., 2024). As shown in Table 10, our models outperform others in in-context learning. In Section A.7, we will further introduce MM1.5-Video, a model variant specifically designed for video understanding.

## 4 CONCLUSION AND LIMITATION

In this work, we build on the insights of MM1 (McKinzie et al., 2024) and introduce MM1.5, a family of highly performant generalist MLLMs. While MM1 focused on key pre-training choices, we improve post-pre-training techniques like continual pre-training, dynamic high-resolution image processing, and curation of our supervised fine-tuning datasets. Our extensive ablations show MM1.5's strong performance in text-rich image understanding, visual grounding, and multi-image reasoning. Like most MLLMs, MM1.5 may produce harmful and counterfactual responses. Future efforts will aim to unify these capabilities for a stronger generalist model, with the hope of benefiting the community beyond specific architectures.

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

## A APPENDIX

### A.1 RELATED WORK

Multimodal Large Language Models (MLLMs) (OpenAI, 2024; Islam & Moushi, 2024; Team et al., 2023; Li et al., 2024d; Huang et al., 2023) have recently emerged as a significant area of research focus. The development of MLLMs can be traced back to Frozen (Tsimpoukelli et al., 2021) and Flamingo (Alayrac et al., 2022; Awadalla et al., 2023), with more recent advancements such as LLaVA (Liu et al., 2023b) and MiniGPT-4 (Zhu et al., 2023) introducing the concept of visual instruction tuning. The past year has witnessed a boom of open-source MLLMs, some of which claim to rival GPT-4o on certain benchmarks. Notable examples include Emu2 (Sun et al., 2023b; 2024), VILA (Lin et al., 2024b), Idefics2/3 (Laurençon et al., 2024a; Laurençon et al., 2024a), Cambrian-1 (Tong et al., 2024a), InternLM-XComposer-2.5 (Dong et al., 2024a; Zhang et al., 2024b), InternVL2 (Chen et al., 2024c;b), MiniCPM-V (Yao et al., 2024b), CogVLM2 (Wang et al., 2023a; Hong et al., 2024a), BLIP-3 (Li et al., 2023d; Xue et al., 2024), LLaVA-OneVision (Li et al., 2024e), Llama3.1-V (Dubey et al., 2024), and the latest Qwen2-VL (Bai et al., 2023).

Research in MLLMs has expanded across several fronts: ($i$) scaling up the pre-training data (Lin et al., 2024b; McKinzie et al., 2024; Xue et al., 2024; Awadalla et al., 2024; Li et al., 2024j) and supervised instruction-tuning data (Hu et al., 2024b; Tang et al., 2024; Laurençon et al., 2024a; Tong et al., 2024a); ($ii$) enhancing high-resolution image comprehension (Lin et al., 2023b; Li et al., 2023i; Liu et al., 2024a; Dong et al., 2024a; Gao et al., 2024; Ge et al., 2024; Chen et al., 2024a; Zhang et al., 2024d; Xu et al., 2024c; Li et al., 2024l); ($iii$) exploring various vision encoders (Tong et al., 2024b; Shi et al., 2024) and vision-language connectors (Cha et al., 2024; Yao et al., 2024a; Li et al., 2024k; Cai et al., 2024); ($iv$) using mixture-of-experts (Lin et al., 2024a; Li et al., 2024g); ($v$) extending LLaVA-like architectures to region-level (Wang et al., 2023b; Zhao et al., 2023; Zang et al., 2023; Peng et al., 2023; Chen et al., 2023b; Zhang et al., 2023; You et al., 2023; Zhang et al., 2024a) and pixel-level (Lai et al., 2024; Rasheed et al., 2024; Yuan et al., 2024; Ren et al., 2024) understanding, multi-image reasoning (Jiang et al., 2024; Li et al., 2024e), UI comprehension (You et al., 2024; Hong et al., 2024b), and video understanding (Lin et al., 2023a; Xu et al., 2024a;b), among others.

Among the extensive body of literature on MLLMs, MM1.5 distinguishes itself as a significant upgrade over its predecessor, MM1 (McKinzie et al., 2024). The MM1.5 model family integrates a diverse set of core capabilities, including text-rich image understanding, visual referring and grounding, and multi-image reasoning. In contrast, recent general-purpose MLLMs such as Cambrian-1 (Tong et al., 2024a) and LLaVA-OneVision (Li et al., 2024e) have shown less satisfactory performance in handling referring and grounding tasks, and GPT-4o has to rely on set-of-mark (SoM) prompting (Yang et al., 2023) to understand image regions.

While several recent works have open-sourced detailed SFT data mixtures for public use (Laurençon et al., 2024a; Tong et al., 2024a), the precise impact of each data category and the best recipe to combine them remain under-explored. This is particularly true for models requiring diverse capabilities. MM1.5 stands out by providing a comprehensive empirical study that presents mature recipes for building performant MLLMs. The extension of MM1.5 to mobile UI understanding further enhances the uniqueness of this work.

Another emerging trend in the field is the development of lightweight MLLMs for potential edge deployment (Jin et al., 2024b; Huang et al., 2024; Beyer et al., 2024; Lu et al., 2024; Hinck et al., 2024; Li et al., 2024l; Zhou et al., 2024; He et al., 2024). In MM1.5, models with 1B and 3B parameters are offered, which outperform similar-sized models, such as Phi-3-Vision (Abdin et al., 2024b) and MiniCPM-V (Yao et al., 2024b).

### A.2 PRE-TRAINING ABLATIONS

Beyond the SFT and continual pre-training, we emphasize the importance of large-scale, task-specific data used during pre-training in establishing robust foundations for models to effectively handle diverse tasks. For knowledge-heavy benchmarks like MMMU (Yue et al., 2023a), we found that model performance is highly sensitive to its text comprehension capabilities. The LLM's ability to understand and process textual content is pivotal in addressing the complex reasoning and knowledge-representation challenges posed by these benchmarks, as also observed in Cambrian-1 (Tong et al., 2024a).

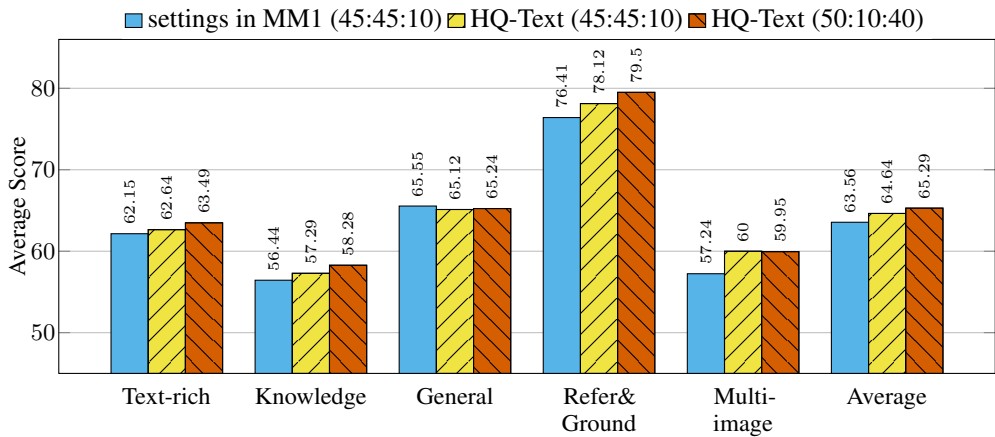

Figure 7: Performance comparison of all categories across different text-only data and pre-training data ratio. The figure highlights the performance improvement when replacing with *HQ-Text* data and the additional gains achieved by adjusting the ratio to 50:10:40. Note that the default setting for continual pre-training (OCR) and *All Mixture* for SFT are used for all models.

We incorporated a higher-quality and more diverse set of text-only datasets, referred to as *HQ-Text*, introduced by (Gunter et al., 2024), during the pre-training phase. These datasets were specifically curated to enhance the model's language capabilities by providing deeper and more varied textual contexts, with a focus on general knowledge, mathematics, and coding. This update aims to strengthen the model's ability in language-based reasoning.

As shown in Figure 7, by simply replacing with the new data, the average score on knowledge improves by 0.85 points.

In conjunction with the text-only datasets and the latest SFT recipes discussed in Section 2.2, we further refined our pre-training data composition. The original data ratio proposed in MM1 (McKinzie et al., 2024) was 45:45:10 for image-caption, interleaved image-text, and text-only data, respectively. Further experiments revealed that decreasing the amount of interleaved pre-training data and, respectively, increasing the weight of text-only data to a ratio of 50:10:40 resulted in improved performance across most tasks after SFT. We note that in contrast to pre-training ablations in MM1, for MM1.5, we conduct evaluations on downstream benchmarks post SFT to select our final pre-training mixture. We hypothesize that relying primarily on few-shot pre-training metrics may not be ideal, as the improvements on such evaluations may not effectively transfer to downstream performance. Our newly optimized data mix for MM1.5 not only enhances multimodal capabilities but also strengthens language understanding, leading to superior overall performance across benchmarks.

With the updated mixture, performance on text-rich average increased by 0.85, knowledge average by 0.99, and refer&ground tasks by around 1.4, as shown in Figure 7. Although there was a slight decrease of 0.05 on multi-image datasets due to the lower weighting of interleaved data, we consider this trade-off reasonable for maintaining strong performance across all tasks.

## A.3    DETAILS OF SFT DATA FOR ABLATION & FINAL SFT MIXTURE

As presented in Table 5, we use a subset of our final SFT data when conducting the ablation study. The MM1.5 final mixture is presented in Figure 8.

## A.4    MM1.5 BENCHMARK DETAILS

Benchmarks used for MM1.5 evaluation are summarized in Table 6, where for each *Category Average Score*, we directly calculate the average of each metric number within that capability category as follows, with detailed evaluation metrics listed in Table 6.

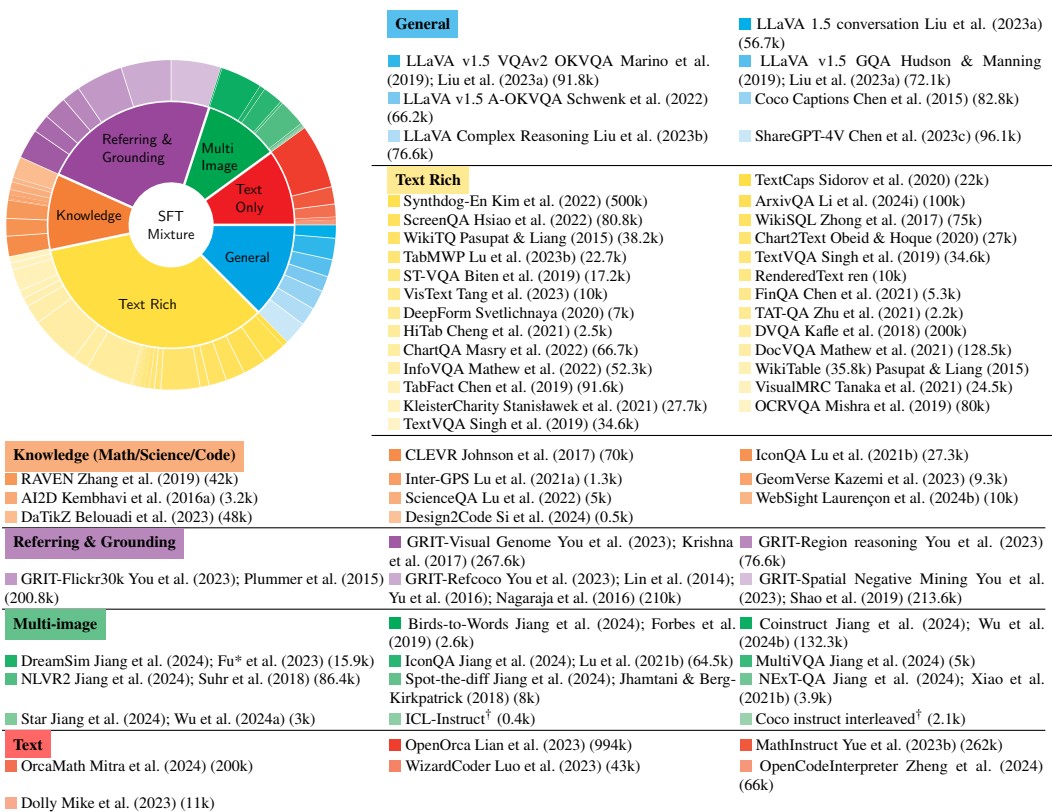

Figure 8: A high-quality data mixture used for MM1.5 supervised fine-tuning, including ($i$) single-image data for enhanced math/science reasoning, text-rich image understanding, and visual referring and grounding, ($ii$) multi-image data, and ($iii$) text-only data. (†) denotes in-house datasets with curation details in Appendix A.9.

Table 5: Overview of the SFT data used in ablation study.

| Data category | Sub-category | Datasets | # QA |
|---|---|---|---|
| **Single-image** | **General** | LLaVA Complex Reasoning, LLaVA Conversation, ShareGPT-4v, Coco Caption, LLaVA v1.5 VQAv2 OKVQA, LLaVA v1.5 GQA, LLaVA v1.5 A-OKVQA | 542K |
| | **Text Rich** | OCRVQA, Synthdog-En, TextCaps, TextVQA, DVQA, ChartQA, DocVQA, InfoVQA, VisualMRC, WikiTQ, DeepForm, KleisterCharity, TabFact | 1.3M |
| | **Refer&Ground** | GRIT-Visual Genome, GRIT-Region reasoning, GRIT-Flickr30k, GRIT-Refcoco, GRIT-Spatial Negative Mining | 1.08M |
| | **Science** | AI2D, ScienceQA | 8K |
| | **Math** | GeomVerse, CLEVER, IconQA, RAVEN, Inter-GPS | 150K |
| | **Code** | WebSight, DaTikZ, Design2Code | 58K |
| **Multi-image** | – | DreamSim, NLVR2, Star, Birds-to-Words, IconQA, Spot-the-diff, ICL-instruct, Coinstruct, MultiVQA, NExT-QA, Coco Instruct Interleaved | 324K |
| **Text-only** | – | Dolly, OpenOrca, MathInstruct, WizardCoder, OrcaMath, OpenCodeInterpreter | 1.71M |

- **General Average Score**: average score of the corresponding metric scores from MME-Normalize[3], Seed-IMG, POPE, LLaVA$^W$, MM-Vet and RealWorldQA.

- **Text-rich Average Score**: average score of the corresponding metric scores from WTQ, TabFact, OCRBench[4], ChartQA[5], TextVQA, DocVQA and InfoVQA.

---

[3]MME-Normalize is (MME-Perception + MME-Cognition)/2800 ×100%.

[4]The accuracy of OCRbench is the total score normalized by 1000 ×100%.

[5]Average of human part accuracy and augmented part accuracy.

Table 6: Details of benchmarks and their metrics used in MM1.5 ablation study. Benchmarks marked with (†) are excluded from the category average.

| Category | Benchmark | Metric |
|---|---|---|
| **General** | MME Fu et al. (2024a) | Normalized Accuracy |
| | SEED Li et al. (2023b) | Seed-IMG |
| | POPE Li et al. (2023h) | Average of random, popular and adversarial |
| | LLaVA-Bench (Wild) Liu et al. (2023b) | GPT-assisted score |
| | MM-Vet Yu et al. (2023) | GPT-assisted score |
| | RealWorldQA x.ai | Accuracy |
| **Text-rich** | WTQ Pasupat & Liang (2015) | Accuracy |
| | TabFact Chen et al. (2019) | Accuracy |
| | OCRBench Liu et al. (2024b) | Accuracy |
| | ChartQA Masry et al. (2022) | Accuracy |
| | TextVQA Singh et al. (2019) | VQA Open Flamingo Accuracy |
| | DocVQA Mathew et al. (2021) | ANLS Score |
| | InfoVQA Mathew et al. (2022) | ANLS Score |
| **Refer&Ground** | Flickr30K Young et al. (2014) | Recall (IoU>0.5, any protocol) |
| | LVIS_Ferret Gupta et al. (2019); You et al. (2023) | Accuracy |
| | Refcoco Kazemzadeh et al. (2014) | Recall@1 (IoU>0.5) |
| | Refcoco+ Kazemzadeh et al. (2014) | Recall@1 (IoU>0.5) |
| | Refcocog Kazemzadeh et al. (2014) | Recall@1 (IoU>0.5) |
| | Ferret-Bench† You et al. (2023) | GPT-assisted score |
| **Knowledge (Math/Science/Code)** | AI2D Kembhavi et al. (2016b) | Accuracy |
| | ScienceQA Lu et al. (2022) | Accuracy-IMG |
| | MathVista Lu et al. (2023a) | GPT-assisted score |
| | MMMU Yue et al. (2023a) | Accuracy |
| **Multi-image** | Qbench2 Zhang et al. (2024e) | Accuracy |
| | Mantis Jiang et al. (2024) | Accuracy |
| | NLVR2 Suhr et al. (2018) | Accuracy |
| | BLINK Fu et al. (2024c) | Accuracy |
| | MVbench Li et al. (2023f) | Accuracy |
| | Muirbench† Wang et al. (2024) | Accuracy |

- **Refer & ground Average Score**: average of the scores of Flickr30k, RefCOCO avg. and LVIS avg., where RefCOCO avg. is the average of RefCOCO A, RefCOCO B, RefCOCO+ A, RefCOCO+ B and RefCOCOg, and LVIS avg. is the average of point and box metrics.

- **Knowledge Average Score**: average score of the corresponding metric scores from AI2D, ScienceQA, MathVista and MMMU.

- **Multi-image average score**: average of Qbench, Mantis, NLVR2, BLINK and MVBench metric scores.

- **MMBaseScore**: average score of the *General Average Score*, *Text-rich Average Score* and *Knowledge Average Score*. This aggregated metric is used in Section 2 to measure the impact of the general, text-rich, and knowledge capabilities of a model.

## A.5 IMPACT OF SYNTHETIC CAPTIONS BY SELF-TRAINING

Besides using public captioning data mentioned in Section 2.3, we also follow (Fang et al., 2024) to study the effect of self-training using synthetic captions. This is particularly important because captioning data generated by black-box commercial models can sometimes be difficult to scale and data from open models, such as from the LLaVA-NeXT family may be constrained by these model's inherent limitations. We develop a self-augmented image-caption data engine building on our previous work, MM1 (McKinzie et al., 2024) with a goal to provide high quality captions in a computationally efficient manner. Specifically, we fine-tune a pre-trained 3B MM1-style model on a mix of synthetic and approximately 8k human-annotated paragraph-length image captions (approximately 70 tokens on average). We then apply this captioner at scale to 290 million web-crawled images with resolutions ranging from 512 to 1024px. Following the approach in (Liu et al., 2024a), we perform concept filtering based on the generated captions, resulting in a dataset of 7 million high-quality captions, which includes numerous text-rich examples and alt-text-derived knowledge.

Building on this, we investigate the impact of the volume of synthetic caption data. The synthetic captions, ranging from 1.4 million to 7 million, show consistent improvements in model performance across various metrics, as illustrated in Figure 9. Specifically, a ratio of 0 on the x-axis indicates that no image-caption data is added to the original OCR dataset, while ratios ranging from 0.2 to 1.0 on the x-axis represent the proportion of the total 7M dataset included in the training. For example, a ratio of 0.2 corresponds to approximately 1.4M image-caption pairs added with the original OCR data while 1.0 denotes all 7M are added into training.

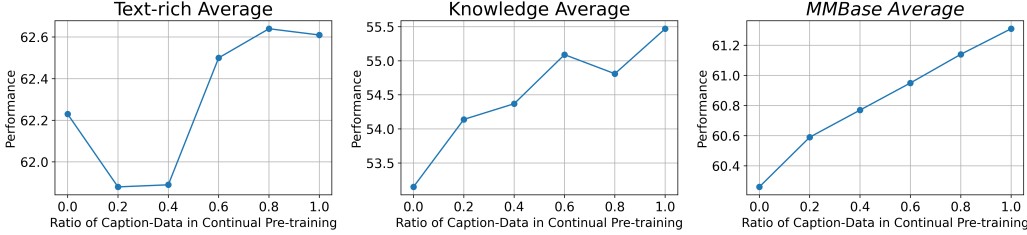

Figure 9: Impact of synthetic captions for continual pre-training the 3B model, building on top of the final MM1.5 strategy introduced in the main text, *i.e.*, including the OCR continual pre-training stage. We report the impact of adding incrementally more synthetic captions, up to 7M in total.

In contrast to the ShareGPT4V-PT and LLaVA-Recap-3M captions as explored in Section 2.3, we find that adding our in-house synthetic captions to the OCR data mixture can lead to consistent improvement for continual pre-training.[6] We observe improvements in knowledge-related benchmarks and the aggregated *MMBase* scores that further scale with increased data volume. This is especially notable since our in-house captioner uses only a 3B model while LLaVA-Recap, for example, uses a 34B model for captioning.

Our results suggest that while a comparatively simple OCR mixture represents a strong baseline for continual pre-training data, high-quality captions can still lead to further improvements. However, the quality, distribution, perhaps even style and length of the generated captions seem crucial to realize gains. While our in-house captions empirically outperformed publicly available data in our specific setting, further research is necessary as to what, specifically, these improvements are attributable to and whether further improvements can be achieved. This investigation goes beyond the scope of this paper, and we aim to study synthetic captioning further in future work.

## A.6 COMPARISON WITH SOTA MODELS

We compared our model with selected benchmarks in Section 3. Here, we present comparisons with more baselines on extensive benchmarks in Table 7 (Knowledge & General), 8 (Text-rich), Table 9 (Refer & s Ground), Table 10 (In-context Learning), and Table 11 (Multi-image).

---

[6]Experiments in this section are based on the final recipe from Section 3, with slightly different settings compared to those in Section A.2.

Table 7: Comparison with SOTA models on knowledge and general benchmarks. (†) The score is the summation of perception and cognition scores. Gemini-1.5-Pro, GPT-4V and GPT-4o numbers are from OpenVLM Leaderboard.

| Model | Knowledge Benchmarks | | | | General Benchmarks | | | | | |
|---|---|---|---|---|---|---|---|---|---|---|
| | AI2D (test) | SQA (test) | MMMU (val) | MathV (testmini) | MME (P/C) | SEED$^I$ | POPE | LLaVA$^W$ | MM-Vet | RealWorldQA |
| *1B Model Comparison* | | | | | | | | | | |
| LLaVAOneVision-0.5B Li et al. (2024c) | 57.1 | 67.2 | 31.4 | 34.8 | 1238.0/240.0 | 65.5 | – | – | 29.1 | 55.6 |
| SPHINX-Tiny Gao et al. (2024) | 24.6 | 21.5 | – | 26.4 | 1261.2/242.1 | – | 82.2 | 52.3 | 23.8 | – |
| DeepSeek-VL Lu et al. (2024) | – | – | 32.2 | 31.1 | – | – | 87.6 | – | 34.8 | – |
| TinyLLaVA Zhou et al. (2024) | – | 60.1 | – | – | – | – | 86.1 | 60.8 | 25.8 | – |
| Gemini Nano-1 Team et al. (2023) | 37.9 | – | 26.3 | 27.3 | – | – | – | – | – | – |
| IntenVL2-2B Chen et al. (2024b) | 74.1 | 94.1 | 36.3 | 46.0 | 1864.3$^†$ | 70.9 | 85.2 | 60.0 | 39.7 | 57.4 |
| MM1-1B McKinzie et al. (2024) | 57.7 | 62.3 | 33.2 | 31.1 | 1393.2/217.1 | 65.6 | 87.4 | 67.5 | 39.4 | 51.2 |
| MM1.5-1B | 59.3 | 82.1 | 35.8 | 37.2 | 1365.7/245.7 | 70.2 | 88.1 | 71.6 | 37.4 | 53.3 |
| MM1.5-1B-MoE | 67.1 | 87.6 | 41.2 | 42.9 | 1511.9/361.1 | 71.4 | 88.6 | 75.5 | 39.8 | 57.8 |
| *3B Model Comparison* | | | | | | | | | | |
| MiniCPM-V 2.0-3B Yao et al. (2024b) | 62.9 | 80.7 | 38.2 | 38.7 | 1808.2$^†$ | 67.1 | 87.8 | 69.2 | 38.2 | 55.8 |
| VILA1.5-3B Lin et al. (2024b) | – | 69.0 | 33.3 | – | 1442.4/– | 67.9 | 85.9 | – | – | – |
| TinyLLaVA Zhou et al. (2024) | – | 69.1 | – | – | 1464.9/– | – | 86.4 | 75.8 | 32.0 | – |
| Gemini Nano-2 Team et al. (2023) | 51.0 | – | 32.6 | 30.6 | – | – | – | – | – | – |
| Bunny He et al. (2024) | – | 78.3 | 41.4 | – | 1581.5/361.1 | 72.5 | 87.2 | – | – | – |
| BLIP-3 Xue et al. (2024) | – | 88.3 | 41.1 | 39.6 | – | 72.2 | 87.0 | – | – | 60.5 |
| Phi-3-Vision-4B Abdin et al. (2024a) | 76.7 | 90.8 | 40.4 | 44.5 | 1441.6/320.0 | 71.8 | 85.8 | 71.6 | 46.2 | 59.4 |
| MM1-3B McKinzie et al. (2024) | 62.4 | 69.4 | 33.9 | 32.0 | 1482.5/279.3 | 68.8 | 87.4 | 72.1 | 43.7 | 55.8 |
| MM1.5-3B | 65.7 | 85.8 | 37.1 | 44.4 | 1478.4/319.6 | 72.4 | 88.1 | 73.0 | 41.0 | 56.9 |
| MM1.5-3B-MoE | 69.9 | 89.8 | 42.9 | 46.9 | 1591.4/365.7 | 73.3 | 87.2 | 76.1 | 43.7 | 60.7 |
| *7B Model Comparison* | | | | | | | | | | |
| LLaVA-NeXT-7B Liu et al. (2024a) | – | 70.1 | 35.8 | 34.6 | 1519.0/332.0 | 70.2 | 86.5 | 81.6 | 43.9 | – |
| Idefics2-8B Laurençon et al. (2024) | – | – | 43.0 | 51.4 | – | – | – | – | – | – |
| MM1-7B McKinzie et al. (2024) | 66.0 | 72.6 | 37.0 | 35.9 | 1529.3/328.9 | 69.9 | 86.6 | 81.5 | 42.1 | 55.7 |
| MM1.5-7B | 72.2 | 89.6 | 41.8 | 47.6 | 1514.9/346.4 | 73.4 | 88.6 | 74.2 | 42.2 | 62.5 |
| *30B Model Comparison* | | | | | | | | | | |
| LLaVA-NeXT-34B Liu et al. (2024a) | – | 81.8 | 51.1 | 46.5 | 1631.0/397.0 | 75.9 | 87.7 | 89.6 | 57.4 | – |
| Cambrian-34B Tong et al. (2024a) | 79.7 | 85.6 | 49.7 | 53.2 | 1689.3/– | 75.3 | – | – | – | 67.8 |
| MM1-30B McKinzie et al. (2024) | 73.3 | 81.0 | 44.7 | 39.4 | 1637.6/431.4 | 72.1 | 87.6 | 89.3 | 48.7 | 59.4 |
| MM1.5-30B | 77.2 | 91.9 | 47.4 | 55.6 | 1646.2/405.7 | 75.0 | 88.6 | 80.4 | 52.0 | 69.0 |
| Gemini-1.5-Pro Reid et al. (2024) | 79.1 | 85.7 | 60.6 | 57.7 | 2110.6$^†$ | – | 88.2 | 95.3 | 64.0 | 64.1 |
| GPT-4V OpenAI (2024) | 75.9 | 82.1 | 53.8 | 48.7 | 1771.5$^†$ | 71.6 | 75.4 | 93.1 | 56.8 | 56.5 |
| GPT-4o Islam & Moushi (2024) | 84.6 | 90.7 | 69.2 | 61.3 | 2310.3$^†$ | 77.1 | 85.6 | 102.0 | 69.1 | 75.4 |

Table 8: Comparison with SOTA models on text-rich benchmarks. Numbers marked with (†) are obtained from Li et al. (2024c).

| Model | WTQ (test) | TabFact (test) | OCRBench (test) | ChartQA (test) | TextVQA (val) | DocVQA (test) | InfoVQA (test) |
|---|---|---|---|---|---|---|---|
| | Text-rich Benchmarks | | | | | | |
| *1B Model Comparison* | | | | | | | |
| LLaVAOneVision-0.5B Li et al. (2024c) | – | – | – | 61.4 | – | 70.0 | 41.8 |
| SPHINX-Tiny Gao et al. (2024) | 15.3 | 51.1 | – | 34.1 | 57.8 | 53.0 | 26.3 |
| DeepSeek-VL Lu et al. (2024) | – | – | 40.9 | – | – | – | – |
| TinyLLaVA Zhou et al. (2024) | – | – | – | – | 51.7 | – | – |
| Gemini Nano-1 Team et al. (2023) | – | – | – | 53.6 | 62.5 | 72.2 | 51.1 |
| InternVL2-2B Chen et al. (2024b) | 35.8 | 56.7 | 78.1 | 76.2 | 73.4 | 86.9 | 58.9 |
| MM1-1B McKinzie et al. (2024) | 19.9 | 49.8 | 56.6 | 61.8 | 68.2 | 68.4 | 38.5 |
| MM1.5-1B | 34.1 | 66.1 | 60.5 | 67.2 | 72.5 | 81.0 | 50.5 |
| MM1.5-1B-MoE | 38.9 | 71.4 | 62.6 | 73.7 | 76.1 | 84.8 | 55.9 |
| *3B Model Comparison* | | | | | | | |
| MiniCPM-V 2.0-3B Yao et al. (2024b) | 24.2 | 58.2 | 60.5 | 59.8 | 74.1 | 71.9 | 37.6 |
| TinyLLaVA Zhou et al. (2024) | – | – | – | – | 59.1 | – | – |
| Gemini Nano-2 Team et al. (2023) | – | – | – | 51.9 | 65.9 | 74.3 | 54.5 |
| BLIP-3-4B Xue et al. (2024) | – | – | – | – | 71.0 | – | – |
| Phi-3-Vision-4B Abdin et al. (2024b) | 47.4 | 67.8 | 63.7 | 81.4 | 70.1 | 83.3 | 49.0 |
| MM1-3B McKinzie et al. (2024) | 23.6 | 52.9 | 57.0 | 66.8 | 71.9 | 75.2 | 44.7 |
| MM1.5-3B | 41.8 | 72.9 | 65.7 | 74.2 | 76.5 | 87.7 | 58.5 |
| MM1.5-3B-MoE | 39.1 | 73.1 | 63.8 | 73.6 | 76.8 | 85.0 | 53.6 |
| *7B Model Comparison* | | | | | | | |
| LLaVA-NeXT-7B Liu et al. (2024a) | – | – | – | – | 64.9 | – | – |
| Idefics2-8B Laurençon et al. (2024a) | – | – | – | – | 73.0 | 74.0 | – |
| DocOwl-1.5-Chat Hu et al. (2024a) | 40.6 | 80.2 | – | 70.2 | 68.6 | 82.2 | 50.7 |
| MM1-7B McKinzie et al. (2024) | 28.8 | 55.5 | 62.6 | 72.6 | 72.80 | 76.8 | 45.5 |
| MM1.5-7B | 46.0 | 75.9 | 63.5 | 78.6 | 76.5 | 88.1 | 59.5 |
| *30B Model Comparison* | | | | | | | |
| LLaVA-NeXT-34B Liu et al. (2024a) | – | – | – | – | 69.5 | – | – |
| Cambrian-34B Tong et al. (2024a) | – | – | 60.0 | 75.6 | 76.7 | 75.5 | – |
| MM1-30B McKinzie et al. (2024) | 33.3 | 58.9 | 60.6 | 76.9 | 73.5 | 75.8 | 47.3 |
| MM1.5-30B | 54.1 | 84.0 | 65.8 | 83.6 | 79.2 | 91.4 | 67.3 |
| Gemini-1.5-Pro Reid et al. (2024) | – | – | 75.4 | 87.2 | 78.7 | 93.1 | 81.0 |
| GPT-4V OpenAI (2024) | – | – | 64.5 | 78.5† | – | 88.4† | – |
| GPT-4o Islam & Moushi (2024) | – | – | 73.6 | 85.7† | – | 92.8† | – |

Table 9: Comparison with SOTA models on referring and grounding benchmarks.

| Model | RefCOCO (testA/B) | RefCOCO+ (testA/B) | RefCOCOg (test) | Flickr30k (test) | LVIS-Ref (box/point) | Ferret-Bench (avg.) |
|---|---|---|---|---|---|---|
| | Refer and Ground Benchmarks | | | | | |
| *1B Model Comparison* | | | | | | |
| SPHINX-Tiny Gao et al. (2024) | 86.9/77.9 | 78.5/63.7 | 78.9 | – | – | – |
| MM1-1B McKinzie et al. (2024) | 0/0 | 0/0 | 0 | 0 | 51.4/51.6 | 47.3 |
| MM1.5-1B | 89.3/81.9 | 83.7/69.3 | 82.8 | 83.0 | 69.7/54.7 | 67.4 |
| MM1.5-1B-MoE | 91.0/84.8 | 86.0/73.0 | 84.7 | 85.4 | 71.4/56.7 | 69.6 |
| *3B Model Comparison* | | | | | | |
| MiniCPM-v2-3B Yao et al. (2024b) | – | – | – | – | 48.2/47.7 | 22.1 |
| Phi-3-Vision-4B Abdin et al. (2024b) | 46.3 / 36.1 | 42.0 / 28.8 | 37.6 | 27.12 | 53.8/54.5 | 32.2 |
| InternVL2 Chen et al. (2024b) | 88.2 / 75.9 | 82.8 / 63.3 | 78.3 | 51.6 | 51.0 / 51.1 | 35.0 |
| MM1-3B McKinzie et al. (2024) | 0/0 | 0/0 | 0 | 0 | 52.9/53.9 | 46.3 |
| MM1.5-3B | 92.0/86.1 | 87.7/75.9 | 86.4 | 85.9 | 76.3/59.5 | 69.5 |
| MM1.5-3B-MoE | 92.6/86.4 | 88.0/77.8 | 86.4 | 85.8 | 79.3/54.5 | 72.2 |
| *7B Model Comparison* | | | | | | |
| Qwen-VL-7B Bai et al. (2023) | 92.3/84.5 | 88.6/76.8 | 86.3 | – | – | – |
| MiniGPT-v2-7B Chen et al. (2023a) | 91.3/84.3 | 85.5/73.3 | 84.3 | – | – | – |
| LLaVA-OneVision-7B Abdin et al. (2024b) | 80.0/61.6 | 76.9/56.2 | 70.0 | 50.1 | 51.2/51.4 | 38.4 |
| Ferret-7B You et al. (2023) | 91.4/82.5 | 87.4/73.1 | 84.8 | 82.2 | 79.4/67.9 | 64.5 |
| Ferret-V2-7B Zhang et al. (2024a) | 94.7/88.7 | 92.8/79.3 | 89.3 | 85.8 | 86.6/74.6 | 75.6 |
| MM1-7B McKinzie et al. (2024) | 0/0 | 0/0 | 0 | 0 | 53.1/53.3 | 48.5 |
| MM1.5-7B | 92.5/86.7 | 88.7/77.8 | 87.1 | 85.3 | 79.4/53.4 | 72.6 |
| *Larger (>13B) Model Comparison* | | | | | | |
| Ferret-13B You et al. (2023) | 92.4/84.4 | 88.1/75.2 | 86.3 | 84.8 | 80.5/68.4 | 66.3 |
| Ferret-V2-13B Zhang et al. (2024a) | 95.0/88.9 | 92.8/81.4 | 90.0 | 86.3 | 87.7/75.1 | 74.9 |
| MM1-30B McKinzie et al. (2024) | 0/0 | 0/0 | 0 | 0 | 53.4/52.7 | 50.9 |
| MM1.5-30B | 94.9/89.5 | 92.4/83.5 | 90.0 | 87.5 | 84.9/61.4 | 77.1 |

Table 10: Comparison with SOTA models on VL-ICL benchmark Zong et al. (2024) for multimodal in-context learning. 4-shot accuracy reported for each subtask.

| Model | VL-ICL Benchmark | | | | | | |
|---|---|---|---|---|---|---|---|
| | CLEVR | Matching MiniImageNet | Open MiniImageNet | Operator induction | Operator induction interleaved | TextOCR | Avg. |
| *1B Model Comparison* | | | | | | | |
| MM1-1B McKinzie et al. (2024) | 25.0 | 49.3 | 73.0 | 16.7 | 8.3 | 33.5 | 34.3 |
| MM1.5-1B | 39.0 | 52.0 | 84.0 | 60.0 | 36.7 | 34.0 | 51.0 |
| MM1.5-1B-MoE | 33.0 | 56.5 | 89.0 | 56.7 | 56.7 | 44.0 | 56.0 |
| *3B Model Comparison* | | | | | | | |
| Phi-3-Vision-4B Abdin et al. (2024b) | 17.0 | 50.0 | 1.0 | 26.7 | 8.3 | 14.0 | 19.5 |
| MM1-3B McKinzie et al. (2024) | 27.5 | 50.0 | 79.0 | 18.3 | 13.3 | 34.0 | 37.0 |
| MM1.5-3B | 33.5 | 59.0 | 88.0 | 48.3 | 66.7 | 42.5 | 56.3 |
| MM1.5-3B-MoE | 32.0 | 58.0 | 92.0 | 63.3 | 65.0 | 47.5 | 59.6 |
| *7B Model Comparison* | | | | | | | |
| OpenFlamingo-9B Awadalla et al. (2023) | 18.8 | 50.0 | 51.2 | 2.8 | 2.8 | 0.0 | 20.9 |
| Idefics-9B Laurençon et al. (2024b) | 27.7 | 50.0 | 53.8 | 7.8 | 6.1 | 22.8 | 28.0 |
| Otter-9B Li et al. (2023a) | 8.2 | 50.4 | 28.5 | 12.2 | 7.2 | 0.8 | 17.9 |
| InternLM-XComposer2-7B Dong et al. (2024a) | 20.0 | 50.1 | 49.0 | 39.4 | 11.1 | 16.0 | 30.9 |
| Qwen-VL-Chat-7B Bai et al. (2023) | 26.8 | 56.4 | 58.0 | 18.9 | 8.9 | 22.3 | 31.9 |
| LLaVA-NeXT-7B Liu et al. (2024a) | 17.8 | 50.0 | 0.0 | 3.3 | 5.0 | 0.0 | 12.7 |
| MM1-7B McKinzie et al. (2024) | 33.0 | 69.5 | 97.5 | 40.0 | 45.0 | 32.0 | 52.8 |
| MM1.5-7B | 25.5 | 52.8 | 98.5 | 68.3 | 60.0 | 31.0 | 56.0 |
| *Larger (>30B) Model Comparison* | | | | | | | |
| Idefics-80B- Laurençon et al. (2024b) | 31.5 | 50.0 | 52.5 | 21.7 | 28.3 | 29.5 | 35.6 |
| Emu2-Chat-37B Sun et al. (2023a) | 14.8 | 50.0 | 28.2 | 21.7 | 10.0 | 36.5 | 26.9 |
| MM1-30B McKinzie et al. (2024) | 25.0 | 63.0 | 98.5 | 51.7 | 38.3 | 36.0 | 52.1 |
| MM1.5-30B | 46.5 | 66.5 | 100.0 | 65.0 | 80.0 | 44.5 | 77.6 |
| GPT-4V OpenAI (2024) | 42.0 | 81.0 | 56.0 | 92.0 | 74.0 | 50.0 | 65.8 |

Table 11: Comparison with SOTA models on multi-image benchmarks. The result with mark (†) in the row of GPT-4V is from GPT-4o. MVBench Li et al. (2024h) is treated as a multi-image benchmark to test the zero-shot transfer capability of MM1.5 to video understanding tasks.

| Model | Multi-image Benchmarks | | | | | |
|---|---|---|---|---|---|---|
| | QBench2 (val) | Mantis (test) | NLVR2 (val) | MVBench | BLINK (val) | Muirbench (test) |
| *1B Model Comparison* | | | | | | |
| LLaVA-NeXT-Interleave-0.5B Li et al. (2024f) | 52.0 | 45.6 | 67.8 | 45.6 | 39.2 | – |
| LLaVAOneVision-0.5B Li et al. (2024c) | 48.8 | 39.6 | 63.4 | 45.5 | 52.1 | 25.5 |
| MM1-1B McKinzie et al. (2024) | 43.4 | 41.5 | 50.9 | 43.8 | 40.3 | 30.7 |
| MM1.5-1B | 66.4 | 50.7 | 79.0 | 45.8 | 46.3 | 34.7 |
| MM1.5-1B-MoE | 70.9 | 51.2 | 83.2 | 48.3 | 43.7 | 40.9 |
| *3B Model Comparison* | | | | | | |
| BLIP-3-4B Xue et al. (2024) | 75.1 | 56.7 | – | – | 49.7 | – |
| Phi-3-Vision-4B Abdin et al. (2024b) | 56.8 | 47.9 | 53.6 | 46.7 | 44.2 | 38.0 |
| MM1-3B McKinzie et al. (2024) | 41.4 | 45.2 | 51.7 | 44.8 | 41.5 | 28.0 |
| MM1.5-3B | 73.2 | 54.8 | 83.8 | 47.7 | 46.8 | 44.3 |
| MM1.5-3B-MoE | 73.8 | 54.4 | 86.0 | 50.3 | 49.8 | 45.6 |
| *7B Model Comparison* | | | | | | |
| LLaVA-v1.5-7B Liu et al. (2023a) | 49.3 | 31.3 | 53.9 | 36.0 | 37.1 | 23.5 |
| LLaVA-NeXT-Interleave-7B Li et al. (2024f) | 74.2 | 62.7 | 88.8 | 53.1 | 52.6 | 38.9 |
| Idefics2-8B Laurençon et al. (2024a) | 57.0 | 48.9 | 86.9 | 29.7 | 45.2 | 26.1 |
| Mantis-Idefics2-8B Jiang et al. (2024) | 75.2 | 57.1 | 89.7 | 51.4 | 49.1 | 44.5 |
| MM1-7B McKinzie et al. (2024) | 43.6 | 51.6 | 59.9 | 45.3 | 40.0 | 30.4 |
| MM1.5-7B | 73.2 | 57.6 | 86.9 | 48.3 | 48.2 | 49.1 |
| *Larger (>14B) Model Comparison* | | | | | | |
| LLaVA-NeXT-Interleave-14B Li et al. (2024f) | 76.7 | 66.4 | 91.1 | 54.9 | 52.1 | – |
| Emu2-Chat-37B Sun et al. (2023a) | 50.1 | 37.8 | 58.2 | 39.7 | 36.2 | 33.6 |
| MM1-30B McKinzie et al. (2024) | 42.8 | 52.5 | 63.1 | 47.1 | 43.5 | 36.7 |
| MM1.5-30B | 79.3 | 64.6 | 90.6 | 54.0 | 50.2 | 58.2 |
| GPT-4V OpenAI (2024) | 76.5 | 62.7 | 88.8 | 43.5 | 51.1 | 68.0† |

### A.7 MM1.5-VIDEO

The multi-image reasoning capability shown in MM1.5 naturally leads us to develop **MM1.5-Video** for video understanding. It takes a video and an instruction as input and generates the response. For the inputs, we uniformly sample $N$ frames from the video at an arbitrary length and feed them into the model as multi-image inputs without special frame assembly. Due to the token limits, we disable the dynamic image splitting for each frame, and the vision encoder generates the feature maps frame-by-frame independently. Specifically, we sample 24 frames for each video, and each frame is represented by 144 tokens.

We introduce two variants for MM1.5-Video. First, we build MM1.5-Video as a *training-free* model, which is achieved by directly adopting the pre-trained MM1.5 image models to video tasks without being fine-tuned on any video data. This saves a lot of computation resources and demonstrates MM1.5's capability of transferring knowledge to new domains.

Second, we introduce the *supervised fine-tuning (SFT)* model where we fine-tune MM1.5 image models on video instruction-tuning datasets to improve its temporal modeling capability for video tasks. We use a mixture of public video datasets from ShareGPTVideo (Zhang et al., 2024c) (556K), VideoChat2 (Li et al., 2023e) (225K), and ActivityNet-QA (Yu et al., 2019) (31.5K). These datasets contain a variety of videos types, spanning different tasks (*e.g.,* open-ended and multiple choice questions), viewpoints (*e.g.,* first- and third-person views), and lengths (*e.g.,* videos from a few seconds to tens of minutes).

#### A.7.1 BENCHMARKS AND METRICS

We compare our video training-free and SFT models with state-of-the-art methods on multiple video question-answering (VideoQA) tasks and benchmarks.

**Open-Ended Benchmarks** evaluate the performance of a model to answer questions in a free-form style. For this task, we include ActivityNet-QA (Yu et al., 2019) and VCGBench (Maaz et al., 2024). Following prior work (Xu et al., 2024b), we use `GPT-3.5-Turbo-0125` to assess the accuracy and score for the prediction. Considering that the labeled answers of these two datasets are typically short (*e.g.,* one word or phrase), we also evaluate on the LLaVA-Hound (Zhang et al., 2024c), which requires the model to generate more detailed answers. This is useful for assessing performance on tasks involving detailed video understanding. We follow their original setting to report the score from `GPT-3.5-Turbo-0301` and consider a score value $\geq 3$ as correct for accuracy calculation.

**Multiple Choice Benchmarks** require the model to pick the correct answer from multiple choices. For this evaluation, we include VideoMME (Fu et al., 2024b), EgoSchema (Mangalam et al., 2024), NExTQA (Xiao et al., 2021a), and IntentQA (Li et al., 2023c). VideoMME is a comprehensive evaluation dataset containing video from a few seconds to one hour in length. EgoSchema consists of egocentric videos and involves complex long-form temporal understanding and reasoning. NExTQA and IntentQA are collected from the same video source, but IntentQA focuses on predicting intents in daily social activities. For all these datasets, the accuracy of selecting the correct answer from the options is used as the evaluation metric.

#### A.7.2 RESULTS

**Training-free results** are shown in Table 12 and 13. MM1.5-Video demonstrates greater capability on Multiple Choice VideoQA, where MM1.5-Video-3B already outperforms state-of-the-art training-free 7B models on all benchmarks. We also find that MM1.5-Video can follow the instruction to precisely output the predicted option; however, most existing methods (Li et al., 2024h) use structured answer prompts (*e.g,* `"Best Option:("`) to guide their models to generate answers in a desirable format. On the other hand, MM1.5-Video achieves only on-par performance compared to SlowFast-LLaVA on the open-ended benchmarks. We hypothesize that this is because our multi-image SFT datasets contain primarily multiple choice tasks, making such a task formulation most similar to the training data.

**SFT results** are also shown in Table 12 and 13. First, we observe that fine-tuning MM1.5-Video on video datasets can improve its performance on all tasks. Second, on both open-ended and multiple choice benchmarks, our small model, MM1.5-Video-1B, significantly outperforms LLaVAOneVision-

Table 12: Comparison with SOTA models on Open-Ended and Multiple Choice benchmarks.

| Model | Video Data | Open-Ended Benchmarks | | Multiple Choice Benchmarks | | | |
|---|---|---|---|---|---|---|---|
| | | ActivityNet-QA (test) | VCGBench (test) | VideoMME (w/o subs) | EgoSchema (subset) | NExTQA (val) | IntentQA (val) |
| *Training-Free Model Comparison* | | | | | | | |
| DeepStack-L-7B Meng et al. (2024) | ✘ | 49.3 | – | – | 38.4 | 61.0 | – |
| IG-VLM-7B (LLaVA-v1.6) Kim et al. (2024) | ✘ | 54.3 | 3.03 | – | 35.8 | 63.1 | 60.3 |
| SlowFast-LLaVA-7B Xu et al. (2024b) | ✘ | 55.5 | 3.04 | 40.7 | 47.2 | 64.2 | 60.1 |
| MM1.5-Video-1B (Training-free) | ✘ | 46.8 | 2.86 | 45.6 | 45.4 | 70.0 | 67.8 |
| MM1.5-Video-3B (Training-free) | ✘ | 50.9 | 3.04 | 48.4 | 48.4 | 72.8 | 72.7 |
| MM1.5-Video-7B (Training-free) | ✘ | 52.5 | 3.05 | 52.4 | 49.6 | 76.1 | 76.7 |
| *SFT Model Comparison* | | | | | | | |
| VideoChatGPT-7B Maaz et al. (2024) | ✔ | 35.2 | 2.42 | – | – | – | – |
| Video-LLaVA-7B Lin et al. (2023a) | ✔ | 45.3 | 2.84 | 39.9 | – | – | – |
| Vista-LLaMA-7B Ma et al. (2024) | ✔ | 48.3 | – | – | – | 60.7 | – |
| MovieChat+-7B Song et al. (2024) | ✔ | 48.1 | – | – | – | 54.8 | – |
| VideoChat2-7B Li et al. (2024h) | ✔ | 49.1 | 2.98 | – | – | 68.6 | 81.9 |
| Video-LLaMA2-7B Cheng et al. (2024) | ✔ | 50.2 | 3.13 | 47.9 | 51.7 | – | – |
| PLLaVA-7B Xu et al. (2024a) | ✔ | 56.3 | – | – | – | – | – |
| LLaVA-NeXT-Interleave-0.5B Li et al. (2024f) | ✔ | 48.0 | 3.07 | – | – | 59.5 | – |
| LLaVA-NeXT-Interleave-7B Li et al. (2024f) | ✔ | 55.3 | 3.42 | – | – | 78.2 | – |
| LLaVAOneVision-0.5B Li et al. (2024c) | ✔ | 50.5 | 3.12 | 44.0 | 26.8 | 57.2 | – |
| LLaVAOneVision-7B Li et al. (2024c) | ✔ | 56.6 | 3.51 | 58.2 | 60.1 | 79.4 | – |
| MM1.5-Video-1B (SFT) | ✔ | 56.1 | 3.14 | 45.7 | 51.0 | 71.8 | 74.2 |
| MM1.5-Video-3B (SFT) | ✔ | 57.9 | 3.17 | 49.5 | 52.4 | 74.7 | 81.2 |
| MM1.5-Video-7B (SFT) | ✔ | 60.9 | 3.22 | 53.5 | 57.2 | 76.9 | 86.6 |

Table 13: Comparison with SOTA models on LLaVA-Hound benchmarks. (†) indicates the published version released at `https://huggingface.co/ShareGPTVideo/LLaVA-Hound-SFT`.

| Model | In-domain Benchmarks | | | Out-of-domain Benchmarks | | | |
|---|---|---|---|---|---|---|---|
| | ActivityNet-QA | VIDAL-QA | WebVid-QA | MSVD-QA | MSRTT-QA | TGIF-QA | SSV2-QA |
| Video-ChatGPT-7B Maaz et al. (2024) | 34.2 | 29.4 | 38.9 | 34.1 | 25.7 | 31.4 | 19.4 |
| LLaMA-VID-7B Li et al. (2023g) | 36.5 | 30.6 | 37.0 | 34.1 | 25.0 | 27.2 | 22.2 |
| Chat-UniVi-7B Jin et al. (2024a) | 39.4 | 31.4 | 40.1 | 35.6 | 25.9 | 33.2 | 20.6 |
| Video-LLaVA-7B Lin et al. (2023a) | 41.4 | 34.3 | 42.5 | 39.5 | 30.8 | 33.0 | 24.3 |
| LLAVA-HOUND-SFT-7B† | 62.8 | 56.3 | 66.8 | 62.2 | 52.6 | 61.1 | 35.4 |
| MM1.5-Video-1B (Training-free) | 49.0 | 42.6 | 55.8 | 49.8 | 43.3 | 47.6 | 27.2 |
| MM1.5-Video-3B (Training-free) | 51.5 | 45.4 | 58.5 | 51.1 | 46.0 | 49.2 | 28.2 |
| MM1.5-Video-7B (Training-free) | 52.8 | 48.7 | 58.5 | 52.9 | 48.1 | 49.8 | 30.4 |
| MM1.5-Video-1B (SFT) | 65.7 | 60.6 | 68.7 | 65.0 | 55.3 | 64.0 | 34.0 |
| MM1.5-Video-3B (SFT) | 67.8 | 63.4 | 71.1 | 65.2 | 57.2 | 64.9 | 35.2 |
| MM1.5-Video-7B (SFT) | 68.5 | 68.5 | 71.5 | 67.2 | 59.3 | 65.5 | 37.9 |

0.5B (*e.g.,* 24.2% on EgoSchema and 14.6% on NExTQA) and achieves the state-of-the-art results. Third, our 7B model achieves state-of-the-art performance on ActivityNet-QA (*e.g.,* outperforming LLaVAOneVision-7B by 4.3%) and very strong results (mostly runner-up) on other benchmarks by using only public video datasets. We are impressed by the superior results of LLaVAOneVision-7B, especially on long-form video benchmarks such as VideoMME and EgoSchema. We hypothesize this can be due to that (*i*) it is trained on their re-annotated video datasets with better labeling quality, (*ii*) it takes more video frames as inputs (*i.e.,* 32 vs. 24), (*iii*) it uses multiple training stages on joint image and video datasets. We will explore these directions to improve our model in future work. Lastly, MM1.5-Video achieves state-of-the-art performance on the LLaVA-Hound benchmarks, which demonstrates our capability for detailed video understanding.

## A.8    MM1.5-UI

One of the most promising applications of MLLMs that has recently gained popularity is using them to understand and act on user interfaces (UIs) on behalf or alongside users (Hong et al., 2024b; Baechler et al., 2024; You et al., 2024; Li & Li, 2023), which could significantly boost users' productivity and efficiency when interacting with digital devices. This application typically involves providing a model input of: (*i*) an image of the graphical user interface (GUI) of a device (*i.e.*, phone or computer) screen; and (*ii*) instructions on either knowledge *grounded* on certain areas or the entirety of the screen (*e.g.*, Is this element at <x1,x2,y1,y2> clickable?), or asking it to *refer* to certain areas of the screen that fit the questions' criterion (*e.g.*, Where is the text 'login' on the screen?). Beyond referring and grounding abilities, excelling on UI tasks also requires text-rich image understanding ability to

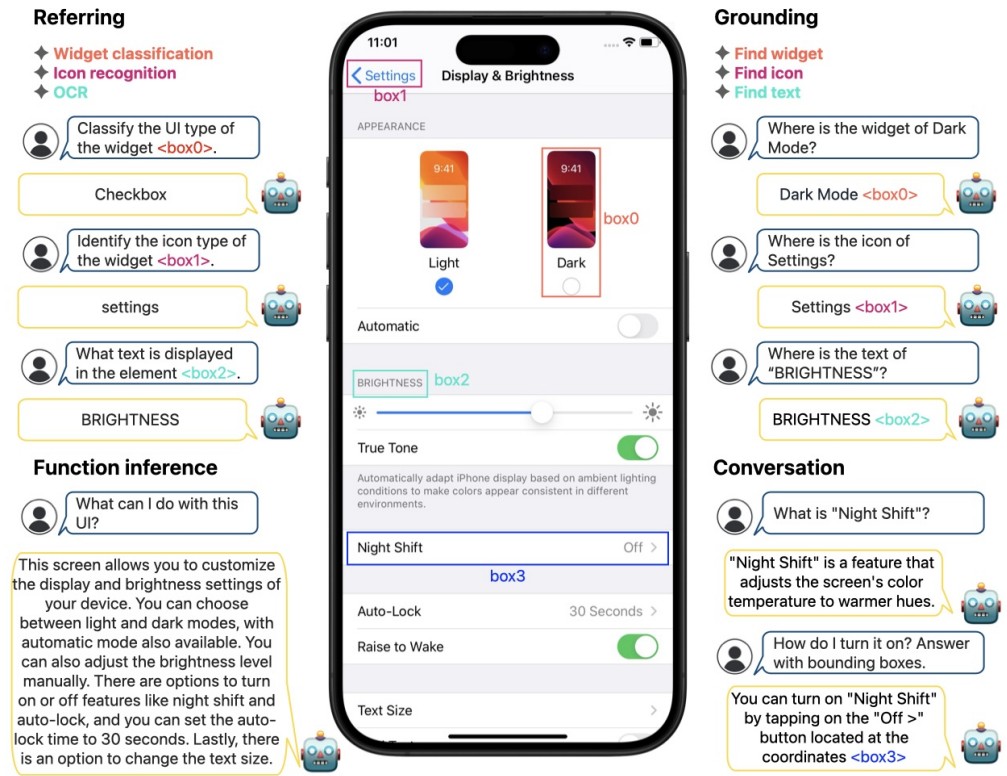

Figure 10: Illustration of the UI understanding capability shown in MM1.5-UI. Our single model is able to perform a variety of referring and grounding tasks and establish new state-of-the-arts. Moreover, it can summarize the functions of the UI screen and engage with users through conversations.

understand text-dense UIs, and background knowledge about typical user interactions on devices, which makes MM1.5 a perfect candidate to be developed into a highly capable UI understanding model.

Towards this goal, we developed **MM1.5-UI**, an MM1.5 model variant further fine-tuned specifically on UI data that achieves competitive performance on UI understanding tasks and establishes new state-of-the-art performance in various benchmarks. Figure 10 illustrates a single MM1.5-UI model's wide range of UI understanding capabilities on an iPhone screenshot. The model can find certain text ("BRIGHTNESS") on the left side (box2), correctly identify the settings icon at the top left (box1), classify a UI element on the right as a checkbox (box0), and maintain a multi-turn conversation about the "Night Shift" function (box3) in the UI.

### A.8.1 BENCHMARKS AND METRICS

We train and evaluate MM1.5-UI on a variety of public and elementary UI understanding tasks used in Ferret-UI (You et al., 2024). These tasks are established benchmarks in literature that cover multiple aspects of UI understanding, and allow us to fairly compare MM1.5-UI against prior work:

- **Public Benchmarks** include screen2words (Wang et al., 2021): a screen-level captioning task; widget captions (Li et al., 2020): a widget-level captioning task; and taperception (Schoop et al., 2022): predicting the tapability of a certain widget on the UI.

- **Ferret-UI** elementary tasks are split into two categories: Grounding (Grd-*) are questions querying for a certain area on the screen, such as finding an icon; and Referring (Ref-*) are questions given a certain area on the screen, such as recognizing text within a screen area (*i.e.*, OCR). Each of these tasks also has an iOS (*-i) and Android (*-A) version, forming four categories of tasks (*e.g.*, Grounding task on Android is Grd-A).

Table 14: Comparison with SOTA models on UI benchmarks. S2W: screen2words, WiC: widget captioning, TaP: taperception. (†) denotes per-task fine-tuning. 1 ep. means 1 epoch model training.

| Model | Public Benchmarks | | | Ferret-UI Elementary Tasks | | | |
|---|---|---|---|---|---|---|---|
| | S2W | WiC | TaP | Ref-i | Ref-A | Grd-i | Grd-A |
| Spotlight Li & Li (2023) | 106.7 | 141.8 | 88.4 | - | - | - | - |
| PaliGemma-3B† Beyer et al. (2024) | 119.6 | 148.4 | - | - | - | - | - |
| Ferret-UI-13B You et al. (2024) | 113.4 | 142.0 | 78.4 | 80.5 | 82.4 | 79.4 | 83.5 |
| MM1.5-UI-1B | 103.0 | 144.4 | 79.3 | 90.0 | 88.6 | 86.5 | 88.2 |
| MM1.5-UI-3B | 103.3 | 145.0 | 80.4 | 90.8 | 89.2 | 87.3 | 88.8 |
| MM1.5-UI-7B | 100.6 | 149.7 | 80.3 | 91.2 | 89.2 | 87.2 | 88.6 |
| MM1.5-UI-30B | 106.0 | 145.9 | 80.6 | 91.8 | 89.7 | 88.2 | 89.1 |
| *Ablation on MM1.5 SFT on UI tasks* | | | | | | | |
| MM1.5-UI-3B (1 ep.) | 103.9 | 145.2 | 77.4 | 88.6 | 87.7 | 86.0 | 87.9 |
| MM1.5-UI-3B (1 ep., w/o MM1.5 SFT) | 103.8 | 139.5 | 75.3 | 88.2 | 87.4 | 85.5 | 87.1 |

More details of the benchmarks can be found in Appendix A.10 and the original Ferret-UI paper (You et al., 2024).

### A.8.2 RESULTS

MM1.5-UI models are trained by further fine-tuning the final MM1.5 models on the Ferret-UI data mixture (You et al., 2024), which includes training data corresponding to the above elementary UI tasks and additional GPT-4-generated conversations about functionalities and descriptions about the UIs' functionality and layouts. There are 801K samples in total. All models are trained with the same batch size and learning rate as the original MM1.5 model.

**Comparison with Prior Art.** Results are summarized in Table 14. Our MM1.5-UI models outperform prior best models in nearly all benchmarks except Screen2words. In particular, even our 1B model is able to outperform the Ferret-UI model in its proposed elementary tasks by a wide margin despite being ten times smaller. The performance difference is most significant on iOS tasks at 9.1 points on average. This demonstrates that the abilities learned by MM1.5 are relevant and useful for UI tasks.

When comparing the performance across individual benchmarks, MM1.5-UI demonstrates a clear hierarchy of difficulties among tasks that focus on different types of UI elements, similar to Ferret-UI (You et al., 2024). Tasks focused on text are the most challenging, followed by those involving icons, while widget-based tasks are the easiest. This trend holds for both referring and grounding tasks. However, MM1.5-UI shows a notable performance improvement in icon-based tasks, significantly narrowing the gap between icon and widget tasks. Ferret-UI highlighted the importance of resolution for tasks involving smaller elements like icons. The higher resolution and dynamic image splitting used in MM1.5-UI further confirm that resolution is particularly beneficial for enhancing performance in icon-related tasks.

**Impact of MM1.5 SFT on UI tasks.** To highlight the effectiveness of the MM1.5 SFT mixture on downstream UI tasks (*i.e.*, in MM1.5-UI), we compare the performance of the full MM1.5-UI model with a baseline UI model fine-tuned with UI data on the pre-training checkpoint that MM1.5 was trained on. Both models are trained for one epoch using the Ferret-UI dataset, and their results are presented in Table 14. The final MM1.5-UI model, which underwent SFT for general domain, text-rich, and refer&ground tasks, achieves superior UI performance within the same number of training steps. This demonstrates the strong transfer capability of MM1.5 for UI applications and contributes to its performance improvement over prior SOTA models.

**Impact of model scaling.** We observe overall performance improvements as models scale, though gains in all metrics remain modest, suggesting that larger models may be constrained by factors such as data diversity, image resolution, or overfitting. For instance, in the most challenging OCR tasks, 47.8% of incorrect responses contain the ground truth as a strict substring of the generated response, or vice versa. This suggests the model accurately recognized the text but failed to trim or include the correct amount. Additionally, the performance of the 7B and 30B models appears to have plateaued,

indicating that larger, more diverse datasets and joint SFT of UI and core capabilities could further improve the performance.

## A.9 MM1.5 IN-HOUSE MULTI-IMAGE SFT DATA

We generated two sets of in-house SFT data for the multi-image capabilities of our model: coco-instruct-interleaved and icl-instruct.

Coco-instruct-interleaved was generated using an LLaVa-style approach using our text-only LLM. We input COCO image captions for two images, along with a prompt requiring the LLM to generate a conversation between a user and an assistant, with the two images inserted in context. This dataset helps our model learn conversational multi-image capabilities. An example from this dataset is shown below. This dataset contains ~2k examples, with 2 images per example.

> **User**: <image 1> What do you see in the following picture?
> **Assistant**: I see an open beige toilet with a monitoring device attached to the seat. The device appears to be displaying various metrics, such as temperature and pressure, and it is likely used for medical purposes.
> **User**: <image 2> What about this picture?
> **Assistant**: I see a row of parking meters sitting next to each other near a street. The meters are typical of those used to enforce parking regulations in many cities.
> **User**: What is the difference between the two pictures?
> **Assistant**: The first picture depicts a toilet with a monitoring device, while the second picture shows a row of parking meters. These are two very different objects in different settings. The first picture is likely used for medical purposes, while the second picture is used to regulate parking in cities.

Icl-instruct was generated manually to help the model retain in-context learning abilities from pre-training along with more nuanced in-context learning tasks that require multi-step reasoning from the model. This dataset contains different varieties of in-context learning tasks, some of which are similar to those found in the VL-ICL benchmark (Zong et al., 2024). We found that including this dataset, which contains ~500 examples, greatly boosts the models' in-context learning performance.

## A.10 MM1.5-UI BENCHMARK DETAILS

The public benchmark tasks and metrics for evaluating MM1.5-UI are:

- **Screen2words** is a captioning task where each complete screen is paired with 5 ground-truth high-level summaries under ten words. The generated summaries' quality is measured by CIDEr score between the ground-truth and generated summaries.

- **Widget Captioning** is a captioning task where a certain screen area that corresponds to a widget (*e.g.*, button, list item) is paired with 3 ground-truth captions. The generated summaries' quality is measured by CIDEr score similar to Screen2words.

- **Taperception** is a binary classification task where a certain screen area that correspond to a widget (*e.g.*, button, list item) is paired with a ground-truth binary label of whether the screen area is 'tappable' (*i.e.*, clickable by users). The generated labels' quality is measured by F1 score.

The Ferret-UI Elementary task benchmarks used to evaluate MM1.5-UI, organized by capability categories, are:

- **Ferret-UI Grounding (Grd-i/A)** is a set of three grounding-based UI tasks introduced in Ferret-UI (You et al., 2024). These tasks query for certain areas of screens that meet certain criteria. They include finding a widget given a text description, finding an icon given the class of the icon, and finding a text location on screen. The expected response from the model is a bounding box, and the quality of the bounding box is measured by Recall with IoU>0.5.

- **Ferret-UI Referring (Ref-i/A)** is a set of three referring-based UI tasks introduced in Ferret-UI (You et al., 2024). These tasks query about knowledge or characteristics that correspond to certain areas of the screens. They include classifying the type of widgets in the given areas, recognizing the type of icons in the given areas, and recognizing texts in the given areas. The quality of the responses is measured by exact match accuracies.

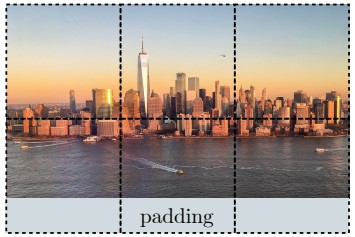 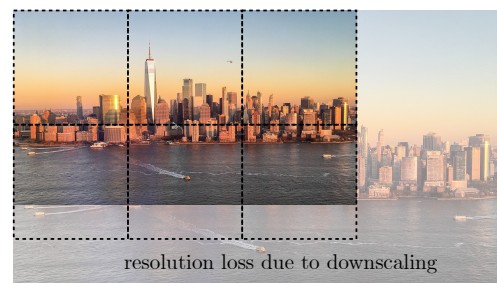

Figure 11: Illustration of image grid selection used in dynamic image splitting for high-resolution image encoding. (Left) If the grid can cover the full image without scaling down, we choose the grid that minimizes padding. (Right) Otherwise, we choose the grid that minimizes the resolution loss due to scaling down.

Each of these two sets Ferret-UI tasks further have two variants with screenshots from two types of operating systems (iOS/Android), of which Android tasks are denoted as -A (*e.g.*, Grd-A), and iPhone tasks as -i, which results in 12 tasks in total.

## A.11 DYNAMIC VS STATIC IMAGE SPLITTING

Detailed ablation study of MM1.5 with different image splitting strategies is shown in Table 15, 16, 17 and 18. All models are using the final setting except for the image splitting (dynamic vs. static).

Table 15: Comparison of our models when using dynamic vs. static image splitting. We follow our final settings for all models. (S) and (D) indicate static and dynamic splitting, respectively.

| Model | Knowledge Benchmarks | | | | General Benchmarks | | | | | |
|---|---|---|---|---|---|---|---|---|---|---|
| | AI2D (test) | SQA (test) | MMMU (val) | MathV (testmini) | MME (P/C) | SEED$^I$ | POPE | LLaVA$^W$ | MM-Vet | RealWorldQA |
| *1B Model Comparison* | | | | | | | | | | |
| MM1.5-1B(S) | 59.5 | 83.9 | 36.1 | 37.4 | 1393.4/244.9 | 69.99 | 87.9 | 67.9 | 34.2 | 51.8 |
| MM1.5-1B(D) | 59.3 | 82.1 | 35.8 | 37.2 | 1365.7/245.7 | 70.2 | 88.1 | 71.6 | 37.4 | 53.3 |
| MM1.5-1B-MoE(S) | 66.4 | 86.8 | 41.2 | 42.2 | 1481.4/293.6 | 71.3 | 89.5 | 76.5 | 41.8 | 60.1 |
| MM1.5-1B-MoE(D) | 67.1 | 87.6 | 41.2 | 42.9 | 1511.9/361.1 | 71.4 | 88.6 | 75.5 | 39.8 | 57.8 |
| *3B Model Comparison* | | | | | | | | | | |
| MM1.5-3B(S) | 66.1 | 87.2 | 36.8 | 43.1 | 1439.8/297.5 | 71.9 | 88.3 | 72.1 | 38.3 | 58.8 |
| MM1.5-3B(D) | 65.7 | 85.8 | 37.1 | 44.4 | 1478.4/319.6 | 72.4 | 88.1 | 73.0 | 41.0 | 56.9 |
| MM1.5-3B-MoE(S) | 66.3 | 89.3 | 41.9 | 43.1 | 1527.8/342.5 | 72.4 | 88.4 | 78.5 | 41.4 | 59.2 |
| MM1.5-3B-MoE(D) | 69.9 | 89.8 | 42.9 | 46.9 | 1591.4/365.7 | 73.3 | 87.2 | 76.1 | 43.7 | 60.7 |
| *7B Model Comparison* | | | | | | | | | | |
| MM1.5-7B(S) | 72.2 | 89.6 | 44.1 | 49.1 | 1531.3/366.4 | 73.5 | 88.6 | 77.2 | 43.3 | 57.0 |
| MM1.5-7B(D) | 72.2 | 89.6 | 41.8 | 47.6 | 1514.9/346.4 | 73.4 | 88.6 | 74.2 | 42.2 | 62.5 |
| *30B Model Comparison* | | | | | | | | | | |
| MM1.5-30B(S) | 75.4 | 92.8 | 46.8 | 56.0 | 1605.2/402.1 | 74.1 | 89.0 | 79.5 | 49.4 | 68.0 |
| MM1.5-30B(D) | 77.2 | 91.9 | 47.4 | 55.6 | 1646.2/405.7 | 75.0 | 88.6 | 80.4 | 52.0 | 69.0 |

## A.12 METHODOLOGY FOR RUNNING COMPETITOR MODELS

This section covers the methodology used to report results for Phi-3-Vision (Abdin et al., 2024b), LLaVA-OneVision (Li et al., 2024c), InternVL2 (Chen et al., 2024b) and MiniCPM-V2 (Yao et al., 2024b). When available, we reported the results published by the original authors, either in their technical reports or on public leaderboards[7]. When not available, we implemented inference runners using publicly released checkpoints. Commonly, we followed (Li et al., 2024b)'s implementations that we adapted on our own internal fork of lm-eval-harness (Gao et al., 2023; McKinzie et al., 2024). To verify the validity of our inference implementations, we ensured we could reproduce previously published results within standard deviation. Below, we share details for each model implementation:

---

[7]https://huggingface.co/spaces/opencompass/open_vlm_leaderboard

Table 16: Comparison of our models when using dynamic vs. static image splitting. We follow our final settings for all models. (S) and (D) indicate static and dynamic splitting, respectively.

| Model | Text-rich Benchmarks | | | | | | |
| | WTQ (test) | TabFact (test) | OCRBench (test) | ChartQA (test) | TextVQA (val) | DocVQA (test) | InfoVQA (test) |
|---|---|---|---|---|---|---|---|
| *1B Model Comparison* | | | | | | | |
| MM1.5-1B(S) | 31.0 | 65.4 | 60.4 | 67.5 | 72.8 | 79.7 | 40.8 |
| MM1.5-1B(D) | 34.1 | 66.1 | 60.5 | 67.2 | 72.5 | 81.0 | 50.5 |
| MM1.5-1B-MoE(S) | 34.1 | 69.6 | 58.0 | 72.7 | 75.8 | 82.5 | 46.0 |
| MM1.5-1B-MoE(D) | 38.9 | 71.4 | 62.6 | 73.7 | 76.1 | 84.8 | 55.9 |
| *3B Model Comparison* | | | | | | | |
| MM1.5-3B(S) | 36.3 | 71.0 | 61.1 | 74.3 | 75.2 | 84.0 | 45.8 |
| MM1.5-3B(D) | 41.8 | 72.9 | 65.7 | 74.2 | 76.5 | 87.7 | 58.5 |
| MM1.5-3B-MoE(S) | 32.5 | 70.1 | 60.2 | 73.0 | 75.9 | 81.0 | 44.2 |
| MM1.5-3B-MoE(D) | 39.1 | 73.1 | 63.8 | 73.6 | 76.8 | 85.0 | 53.6 |
| *7B Model Comparison* | | | | | | | |
| MM1.5-7B(S) | 38.4 | 73.7 | 59.7 | 77.9 | 76.1 | 84.5 | 47.3 |
| MM1.5-7B(D) | 46.0 | 75.9 | 63.5 | 78.6 | 76.5 | 88.1 | 59.5 |
| *30B Model Comparison* | | | | | | | |
| MM1.5-30B(S) | 46.0 | 81.0 | 64.5 | 82.4 | 78.7 | 88.5 | 53.2 |
| MM1.5-30B(D) | 54.1 | 84.0 | 65.8 | 83.6 | 79.2 | 91.4 | 67.3 |

Table 17: Comparison of our models when using dynamic vs. static image splitting. We follow our final settings for all models. (S) and (D) indicate static and dynamic splitting, respectively.

| Model | Refer and Ground Benchmarks | | | |
| | RefCOCO avg. | Flickr30k (test) | LVIS avg. | Ferret-Bench avg. |
|---|---|---|---|---|
| *1B Model Comparison* | | | | |
| MM1.5-1B(S) | 82.0 | 82.7 | 62.4 | 69.7 |
| MM1.5-1B(D) | 81.4 | 83.0 | 62.2 | 67.4 |
| MM1.5-1B-MoE(S) | 79.3 | 80.9 | 63.9 | 73.4 |
| MM1.5-1B-MoE(D) | 84.8 | 85.4 | 64.6 | 69.6 |
| *3B Model Comparison* | | | | |
| MM1.5-3B(S) | 85.1 | 85.3 | 68.1 | 71.2 |
| MM1.5-3B(D) | 85.6 | 85.9 | 67.9 | 69.5 |
| MM1.5-3B-MoE(S) | 82.8 | 82.6 | 67.5 | 70.8 |
| MM1.5-3B-MoE(D) | 86.2 | 85.8 | 66.9 | 72.2 |
| *7B Model Comparison* | | | | |
| MM1.5-7B(S) | 87.2 | 86.0 | 68.8 | 71.2 |
| MM1.5-7B(D) | 86.6 | 85.3 | 66.4 | 72.6 |
| *30B Model Comparison* | | | | |
| MM1.5-30B(S) | 90.1 | 87.7 | 73.2 | 75.6 |
| MM1.5-30B(D) | 90.1 | 87.5 | 73.1 | 77.1 |

**Phi-3-Vision.** We used the public Phi-3-Vision checkpoint[8] and ran it on our families of benchmarks. For **general**, **text-rich**, **knowledge** and **refer&ground** benchmarks, when the position of the image is not determined by the task, we prepend the image to the text input following the examples given in the Phi-3-Vision cook book[9]. For **grounding** benchmarks, we introduced the following prompt: "Question: {question}Answer this question by listing the requested entities and their bounding boxes. The bounding boxes are formatted as follows: <x1,y1,x2,y2>, each value is between 0-{upper_bound}.<n>Answer:". For both **referring** and **grounding**, we experimented with a variety of upper bound bounding boxes. Through our experiments, we noticed that an upper box of 1 yielded to

---

[8] https://huggingface.co/microsoft/Phi-3-vision-128k-instruct
[9] https://github.com/microsoft/Phi-3CookBook/blob/main/md/03.Inference/Vision_Inference.md#3-comparison-of-multiple-images

Table 18: Comparison of our models when using dynamic vs. static image splitting. We follow our final settings for all models. (S) and (D) indicate static and dynamic splitting, respectively.

| Model | Multi-image Benchmarks | | | | | |
|---|---|---|---|---|---|---|
| | QBench2 (val) | Mantis (test) | NLVR2 (val) | MVBench | BLINK (val) | Muirbench (test) |
| *1B Model Comparison* | | | | | | |
| MM1.5-1B(S) | 65.8 | 48.4 | 78.6 | 46.1 | 41.9 | 34.0 |
| MM1.5-1B(D) | 66.4 | 50.7 | 79.0 | 45.8 | 46.3 | 34.7 |
| MM1.5-1B-MoE(S) | 70.2 | 52.1 | 83.0 | 47.4 | 44.8 | 42.5 |
| MM1.5-1B-MoE(D) | 70.9 | 51.2 | 83.2 | 48.3 | 43.7 | 40.9 |
| *3B Model Comparison* | | | | | | |
| MM1.5-3B(S) | 72.0 | 53.5 | 83.9 | 47.8 | 42.5 | 44.5 |
| MM1.5-3B(D) | 73.2 | 54.8 | 83.8 | 47.7 | 46.8 | 44.3 |
| MM1.5-3B-MoE(S) | 70.4 | 54.4 | 85.3 | 47.2 | 47.1 | 44.2 |
| MM1.5-3B-MoE(D) | 73.8 | 54.4 | 86.0 | 50.3 | 49.8 | 45.6 |
| *7B Model Comparison* | | | | | | |
| MM1.5-7B(S) | 73.0 | 56.7 | 87.2 | 49.7 | 47.6 | 53.8 |
| MM1.5-7B(D) | 73.2 | 57.6 | 86.9 | 48.3 | 48.2 | 49.1 |
| *30B Model Comparison* | | | | | | |
| MM1.5-30B(S) | 77.0 | 64.5 | 90.2 | 49.9 | 48.4 | 60.1 |
| MM1.5-30B(D) | 79.3 | 64.6 | 90.6 | 54.0 | 50.2 | 58.2 |

better results, in line with the answers produced by Phi-3-Vision. For Flickr30k, we slightly simplified the benchmark and asked the model to ground one entity per prompt, as grounding multiple entities jointly did not lead to satisfactory results.

**LLaVA-OneVision.** We used the public checkpoint LLaVA-OneVision 7B[10] and we followed closely the LLaVA documentation[11]. When not baked directly into the benchmarks, we used the original LlaVA prompts specified in (Liu et al., 2023a; Li et al., 2024c) for all families of benchmarks. In particular, for **grounding** benchmarks, we used the prompt introduced in Table 18 of (Li et al., 2024c): "Provide the bounding box coordinate of the region this sentence describes". On Flickr30k, we followed the single-entity approach outlined above.

**InternVL2.** The InternVL2 authors provided already a comprehensive set of benchmarks on **general**, **text-rich**, **knowledge** and **refer&ground** families[12], which we reported first. For the few remaining benchmarks, we used the 2B public checkpoint released. InternVL2 code base relies on both VLMEvalKit (Duan et al., 2024) and a custom internal evaluation[13]. We carefully reviewed the logic implemented[14], especially regarding the decoding parameters and prompts used. For **grounding**, we used the prompt shared by the authors: "Please provide the bounding box coordinates of the region this sentence describes: <ref>{question}</ref>". On Flickr30k, we followed the single-entity approach outlined above.

**MiniCPM-V2.** We used the publicly released MiniCPM-V2 2.8B checkpoint[15]. Similarly to InternVL2, MiniCPM-V2 code base relies on both VLMEvalKit and a custom internal implementation[16], which we reviewed carefully to reproduce decoding parameters and prompts. For **refer&ground** benchmarks, we noticed that regardless of the prompt used, MiniCPM-V2 could not produce satisfac-

---

[10] https://huggingface.co/lmms-lab/llava-onevision-qwen2-7b-ov
[11] https://github.com/LLaVA-VL/LLaVA-NeXT/blob/main/docs/LLaVA_OneVision_Tutorials.ipynb
[12] https://huggingface.co/OpenGVLab/InternVL2-2B
[13] https://internvl.readthedocs.io/en/latest/internvl2.0/evaluation.html
[14] https://github.com/OpenGVLab/InternVL
[15] https://huggingface.co/openbmb/MiniCPM-V-2
[16] https://github.com/OpenBMB/MiniCPM-V/tree/main/eval_mm

tory results on RefCOCO and Flickr30k[17]. We decided not to include those results. For **multi-image** and **in-context learning** benchmarks, we found that MiniCPM-V2 does not accept multiple images as input in its Hugging Face implementation. We acknowledge MiniCPM-V2's authors used a custom image concatenating approach for BLINK[18], but we could not reproduce easily the same approach on other benchmarks. We decided to only include the BLINK score reported by the authors[19].

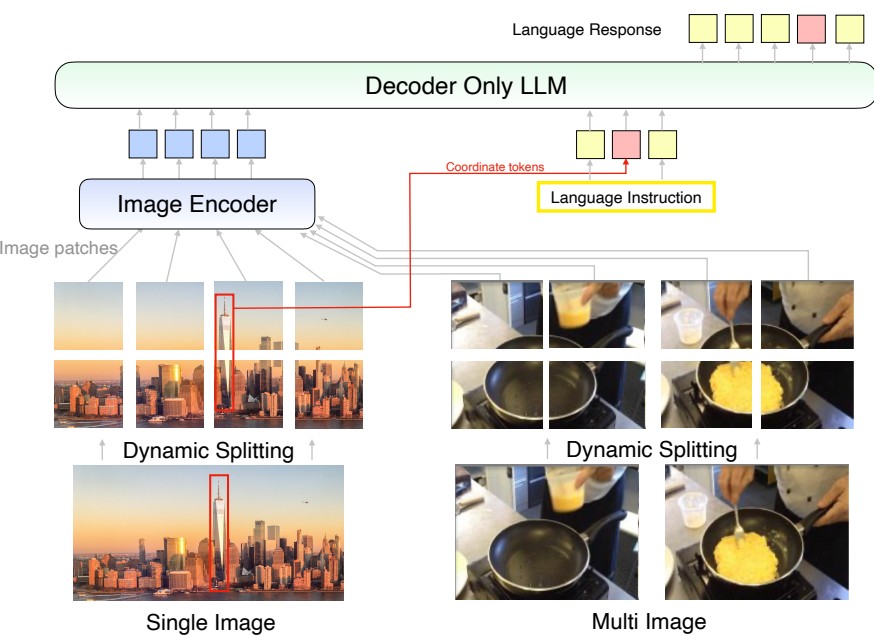

Figure 12: The overview of model architecture. MM1.5 excels at ($i$) text-rich image understanding with dynamic image splitting, ($ii$) visual referring and grounding with coordinate tokens, and ($iii$) multi-image reasoning.

---

[17]Responses commonly produced incomplete bounding boxes, preventing any parsing for a fair evaluation.

[18]https://github.com/OpenBMB/MiniCPM-V/blob/a209258d851f404485e5ae25864417dff3bb74ca/eval_mm/vlmevalkit/vlmeval/api/base.py#L260

[19]https://huggingface.co/spaces/opencompass/open_vlm_leaderboard

## A.13 QUALITATIVE EXAMPLES

### A.13.1 MM1.5 CAPABILITIES

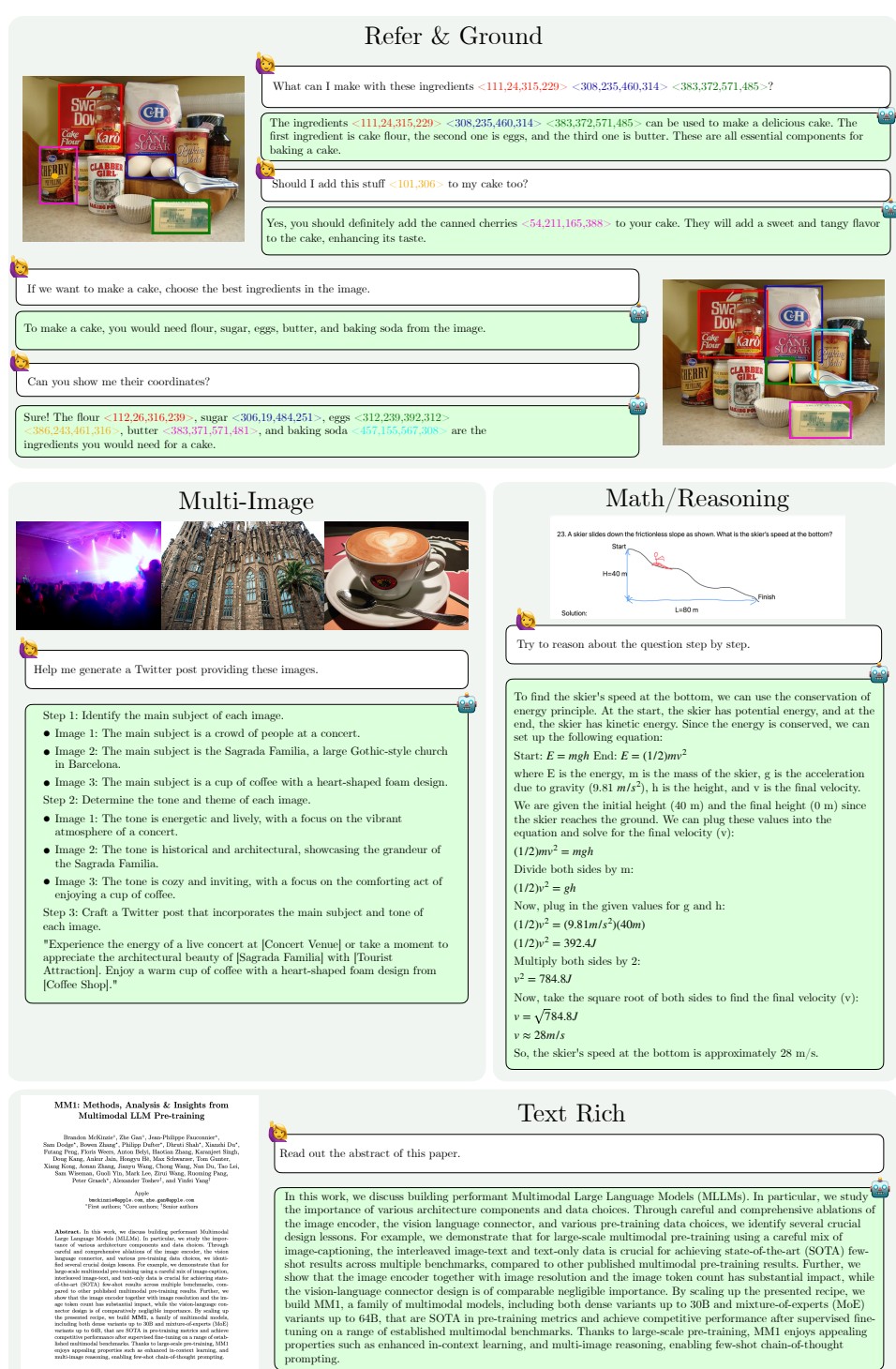

Figure 13: Examples of MM1.5 capabilities. The examples we demonstrated are generated by the MM1.5-3B model..

A.13.2  TEXT-RICH IMAGE UNDERSTANDING

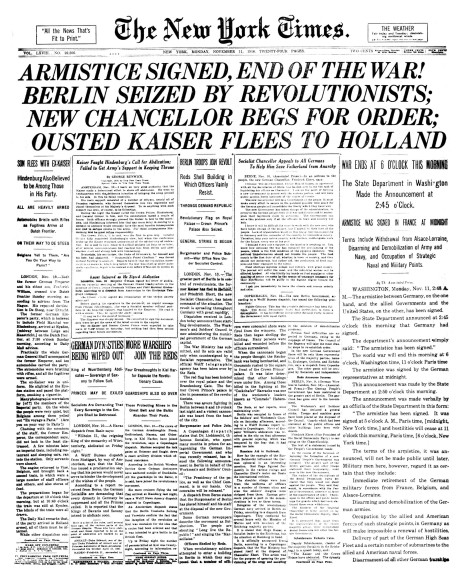

Could you summarize every article on this newspaper page?

The newspaper page is from The New York Times dated November 11, 1918. The main headline announces the signing of the Armistice, marking the end of World War I. Other articles cover the revolution in Germany, with Berlin being seized by revolutionists and the abdication of Kaiser Wilhelm II. The page also includes a call for the abdication of the German Emperor by the socialist Chancellor, Friedrich Ebert. Additionally, there are reports on the ousting of Kaiser Wilhelm II and his flight to the Netherlands, as well as the resignation of the German General Staff. The page also features a report on the armistice being signed in France, marking the end of the war.

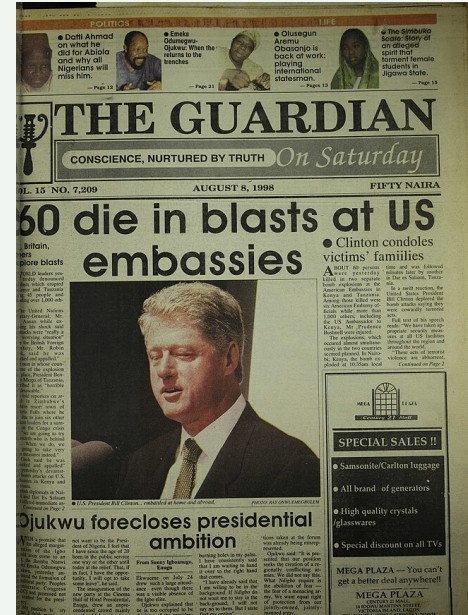

Could you summarize the articles in this newspaper page?

The newspaper page features several articles. The main headline reports on the tragic event of 60 people dying in blasts at US embassies. Another article discusses President Clinton consoling the victims' families. There's also a piece about Ojukwu foreclosing on a presidential ambition. Additionally, the page includes a special sales advertisement for Samsonite/Carlton luggage and high-quality crystals.

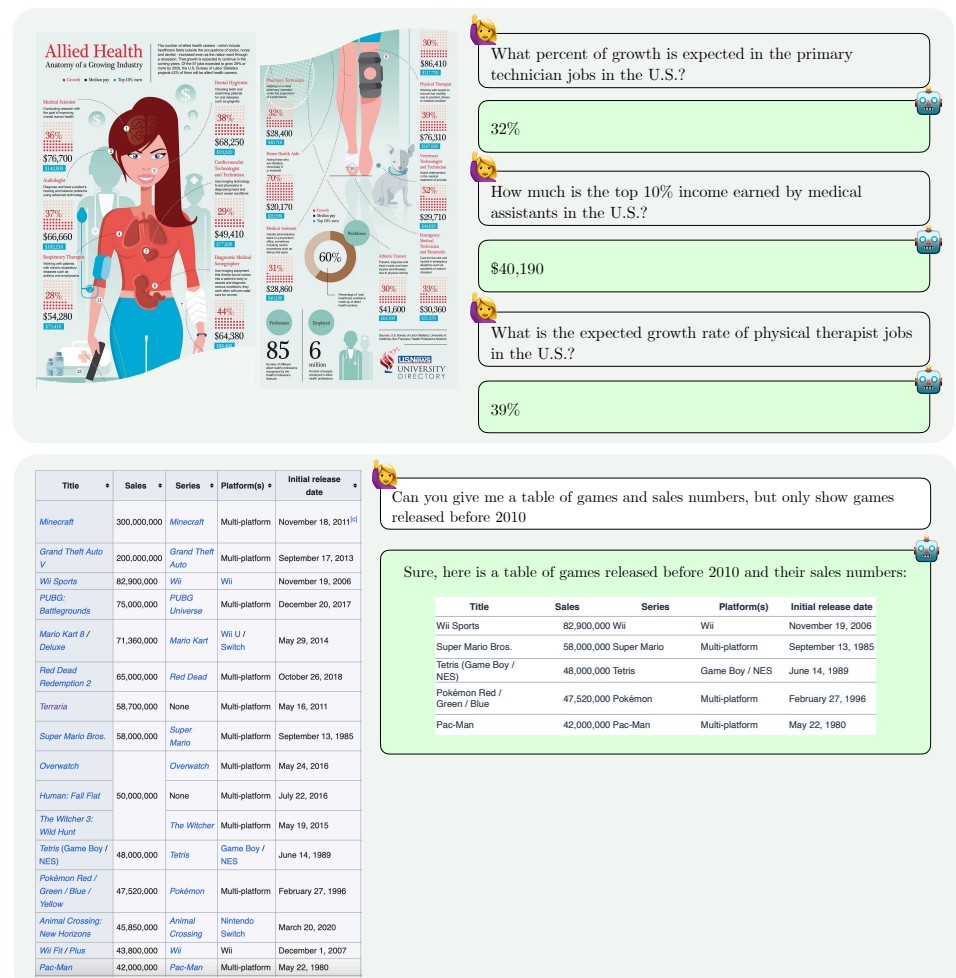

### A.13.3 Visual Referring and Grounding

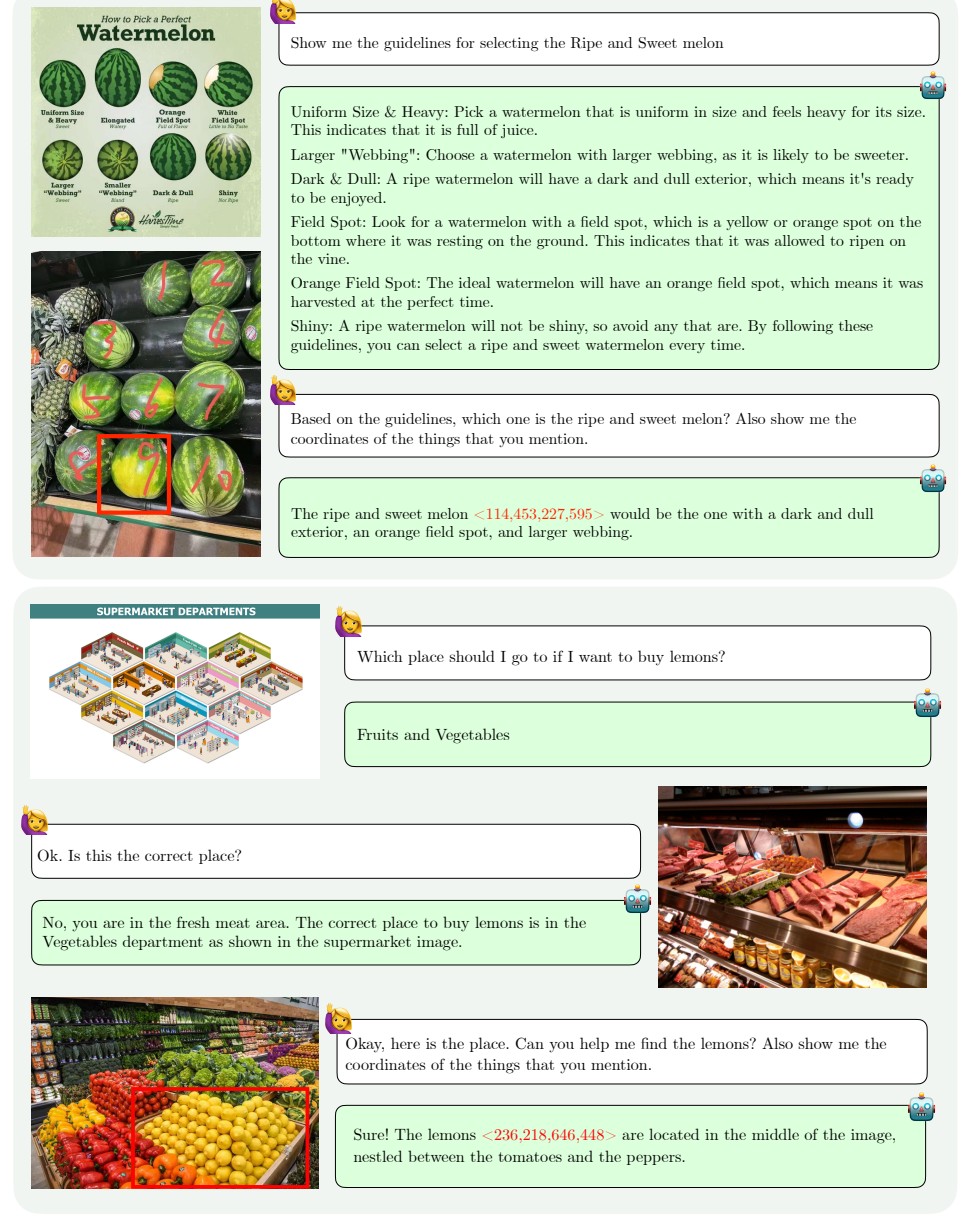

### A.13.4 MULTI-IMAGE REASONING

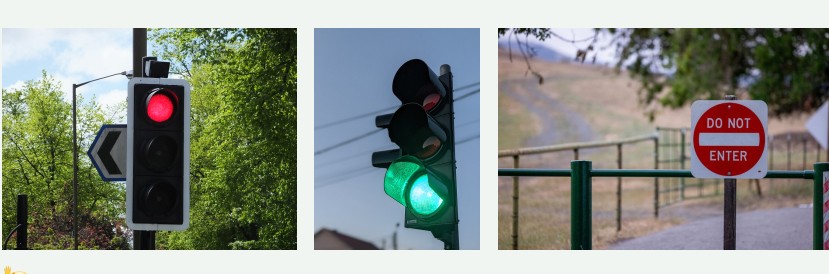

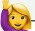 In which of these situations can I drive on ahead?

In the second image, you can drive on ahead because the traffic light is green.

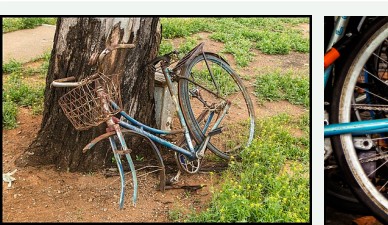 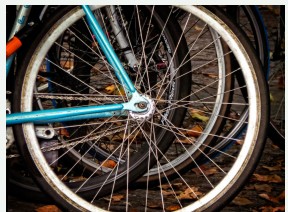 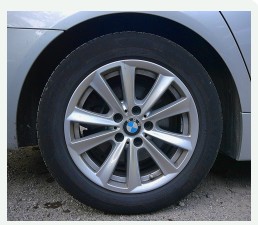

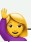 What is the bike experiencing in image 1? Which one fits best, image 2 or image 3, and why?

The bike in image 1 is experiencing rust and wear, as indicated by the rusted metal parts and the bent handlebars. Image 2, with its focus on the bicycle wheel, fits best as it provides a close-up view of the bike's wheel, which is the part most directly related to the bike's condition.

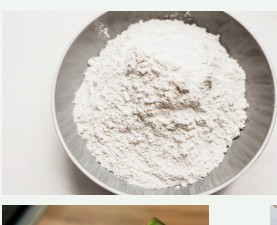 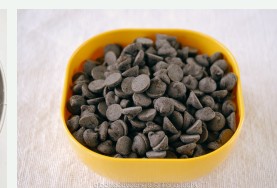

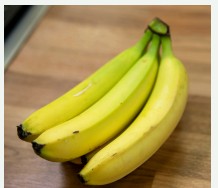 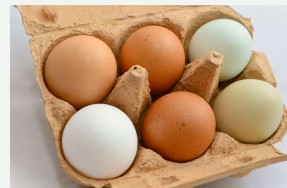

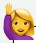 I have these ingredients in my fridge. What can I make using all of these? Explain.

You can make a banana bread using all of these ingredients. The ripe bananas from image 3 can be mashed and combined with the flour from image 1 and the eggs from image 4 to create a moist and flavorful banana bread. The chocolate chips from image 2 can be mixed into the batter for added texture and sweetness.

### A.13.5 UI Understanding

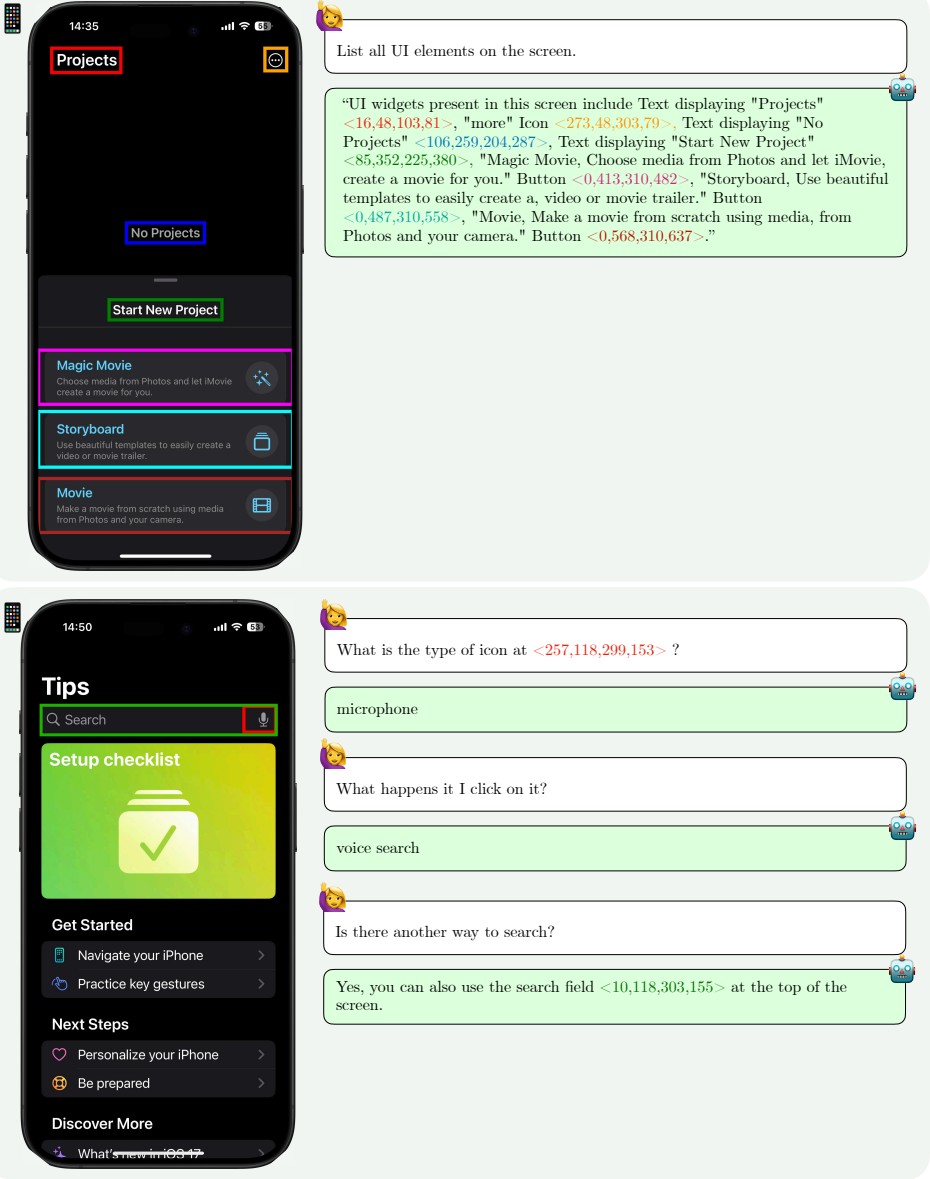

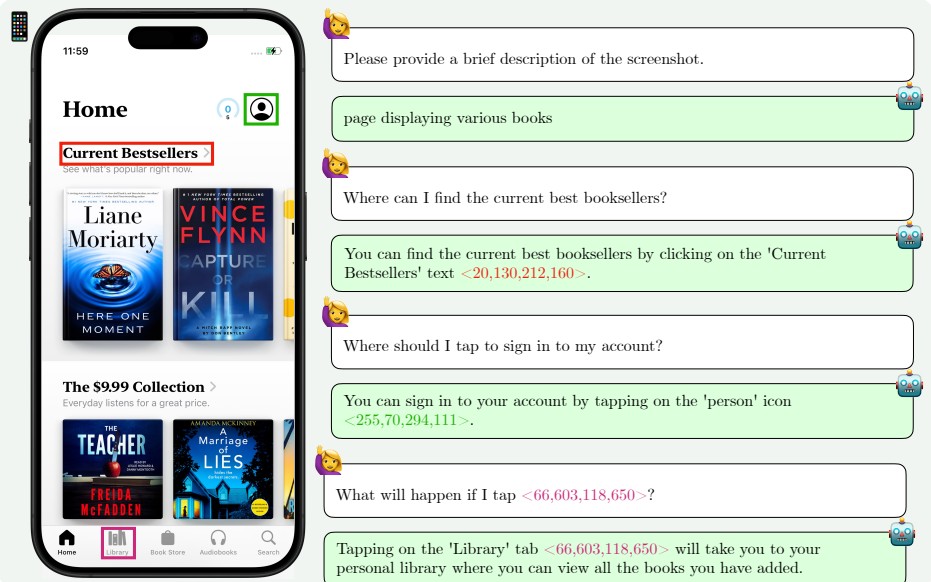

