# OpenReview forum: "MM1.5: Methods, Analysis & Insights from Multimodal LLM Fine-tuning"
_ICLR.cc/2025/Conference — ICLR 2025 Poster_

### Official Review · Reviewer_CjsW · 2024-10-24

**Soundness:** 4
**Presentation:** 3
**Contribution:** 3
**Rating:** 8
**Confidence:** 3

**Summary:**

The paper introduces a new family of multimodal large language models (MLLMs), ranging from 1 to 30B parameters, including MoE models. They build on top of MM1 and focus on data curation for **continual pretraining** and **supervised fine-tuning** (SFT), providing several ablations for the choice of data mixture(s) and summarize their findings in a recipe.

To run their ablations, they use the following setup:

- Static image splitting, splitting large images into four sub-images, plus an overview image, for both SFT and continual pretraining.
- They use either single image or multi image data, and enable splitting only when processing less than three images for a single sample.
- They use a custom (non-disclosed) CLIP text encoder and LLM as their backbones.

For evaluation, they consider general, text-rich, refer & ground, knowledge, and multi-image tasks and report two metrics:

- Category average score: average score for all benchmarks in a given category.
- MMBase score: average score on general, text-rich, and knowledge categories.

## Supervised fine-tuning

SFT is conducted using a large amount of datasets, categorized as follows:

- Single image
	- General
	- Text rich
	- Refer & Ground
	- Science
	- Math
	- Code
- Multi-image
- Text only

They first focus on the impact of single data categories, finding that text-rich data is crucial for text-rich tasks and that good refer & ground capabilities only emerge if the model is trained on appropriate data.

They then focus on optimising for the sampling ratio between the general category and other categories (for single image data), and find that the right mixture of science and math data can significantly improve performance. They also find that doubling the sampling probability of refer & ground data provides a good tradeoff between general performance and refer & ground performance.

Once the single-image data ratios are fixed, they integrate text-only and multi-image data. They find that text-only data has a minor impact and fix its ratio to 0.1. They pick a similar ratio for multi-image data, as it improves the multi-image score significantly while also slightly improving the MMBase score. A larger quantity of multi-image samples decreases the MMBase score. Finally, they fix the single-image ratio to 0.8.

They combine these results to produce three data mixtures:

- Base: general, text-rich, science, math, code.
- Single image: base + refer & ground.
- All: single image + multi-image and text only.

As expected from the previous findings, the single-image mixture unlocks the refer & ground capabilities, while the multi-image one largely improves the multi-image capabilities. The **all** mixture is the best on average (although the base mixture performs best on tasks that are not refer & ground and multi-image).

## Continual pretraining

The paper explores continual pretraining using ~50M OCR high-resolution (1344 x 1344) sampled across four datasets. Once continual pretraining is complete, they perform SFT using the Base mixture. Their findings are:

OCR data must be high-res. Downsampling images to the encoder input dimension (378 x 378) can cause the model to lose up to 2% in average score and even perform worse than when no continual pretraining is performed! OCR data improves performance, but only if it has a high enough resolution.

- They also experiment with data with synthetic captions and find that they improve performance compared to a baseline without continual pretraining. Still, they perform worse than only using OCR data, even when combined.
- In the appendix, they show that self-training with synthetic captions can improve performance compared to an OCR-only baseline.

## Dynamic Image Splitting

These experiments use the single image mixture and start from an MM1 checkpoint without continual pretraining.

The authors claim that static image splitting has inherent issues, as low-res images are also split, and splitting non-square images can result in sub-images that consist only of padding. Thus, they experiment with dynamic image splitting, picking a grid that minimizes the amount of padding. In addition to the sub-images, they include the resized original image for additional context and optionally positional indicators for sub-images.

Based on their results, they claim that:

Dynamic image splitting outperforms static splitting for text-rich tasks. Image resolution, the number of sub-images, and the number of image tokens are important. Moreover, it greatly improves performance on tasks whose images have unusual aspect ratios.

- These benefits come at a relatively small cost (< +10%) in terms of encoded images
- Positional indicators are useful for specific tasks but not necessary in general
- Placing the overview image after the sub-images improves performance

## Final Results

Given the findings illustrated in previous sections, the authors propose a three-step recipe (MM1 pretraining, continual pretraining, and SFT) to train 1, 3, 7, and 30B models, plus 1 and 3B MoE models.

Their models improve over similarly-sized models, sometimes by large margins, and the performance scales well as models grow in parameters. MoE models often outperform their dense counterparts. This is particularly evident for 1B models, where the MoE variant often improves by over 5% compared to its dense counterpart. Their models also provide significant improvements in refer & ground and in-context learning tasks.

**Strengths:**

The paper is well-written and provides novel, valuable, and actionable findings for the MLLM community. While the backbones to replicate the experiments are not publicly available, the authors release as many details as possible regarding their datasets, data mixtures, hyper-parameters, and data processing strategies (e.g., dynamic image splitting, positional indicators, and overview image position).

Moreover, the models trained using the data curation strategies illustrated in the paper significantly outperform a model with the same architecture but a simpler data curation method (MM1) and several current state of the art models, especially in refer & ground, multi image and in-context learning tasks, across various sizes.

**Weaknesses:**

While there are clear benefits in using dynamic image splitting for processing input images, I am not fully convinced it leads to better performance overall, as when comparing similar setups in terms of images and tokens in Table 1 static image splitting works best. I am wondering if it would outperform dynamic image splitting when $n = 10$ and 1440 total tokens are processed.

I am also unsure why the final recipe used indices to indicate the positions for sub-images, even though Table 3 shows that separators work better overall. Could you please clarify this choice?

Presentation-wise, I only have a couple of minor remarks:

1. In section 2.2.1 (in the second paragraph), the authors claim that "adding the code category results in a slight increase in the text-rich score" (i.e., ~1.54%), and that "adding text-rich data can significantly improve the performance on [...] and knowledge benchmarks" (i.e., ~1.78%). I would maybe slightly rephrase either of those sentences.
2. Bolding the best results in tables would make it easier to identify the best configuration(s).
3. Maybe introducing the continual pretraining ablations first and following up with supervised fine-tuning would improve readability. The paper would then follow the recipe logically.

**Questions:**

- If I understand correctly, in section 2.2.2, using a data ratio of 1:2, e.g., for general and science categories results in a sample from the general category (68x the size of science) being 34x more likely to be picked compared to a science sample when building a batch, correct?
- Is dynamic image splitting used in both continual pretraining and supervised fine-tuning?
- Why were the 7B models not compared with similarly sized open-source models such as LLava 1.5 7B?

---

> ### Author Response · Authors · 2024-11-22
> **Author Rebuttal (1/2)**
>
> Thank you for the insightful feedback. We will incorporate your suggestions to improve our presentation. Please see our response to your questions below.
>
> >**Q1**: While there are clear benefits in using dynamic image splitting for processing input images, I am not fully convinced it leads to better performance overall, as when comparing similar setups in terms of images and tokens in Table 1 static image splitting works best. I am wondering if it would outperform dynamic image splitting when n=10 and 1440 total tokens are processed.  I am also unsure why the final recipe used indices to indicate the positions for sub-images, even though Table 3 shows that separators work better overall. Could you please clarify this choice?
>
> **A1**: Thank you for these insightful observations. We would like to address each point:
>
> Regarding performance comparison, Table 1 should be interpreted together with the broader ablations in Table 2, which shows that dynamic splitting with (n_min, n_max) = (4, 9) actually achieves better performance than static splitting. For example, dynamic splitting improves text-rich from 57.7 (Table 1, row 2) to 60.0 (Table 2, row 2). The apparent advantage of static splitting in some configurations in Table 1 is primarily due to its constrained setting (fixed n=4), while dynamic splitting shows greater benefits when allowed appropriate flexibility in grid selection. Furthermore, dynamic splitting with (n_min, n_max) = (4, 9) does not significantly increase the number of sub-images processed. We randomly sample 100k examples from our training data. With these examples, static splitting generates 500k sub-images, while dynamic splitting with (n_min, n_max) = (4, 9) produces barely more, only 539k images in total.
>
> Regarding the choice of position indicators, while Table 3 shows marginally better performance with separators on some metrics (e.g., 74.3 vs 73.4 on DocVQA), we chose to use indices in our final recipe for two practical reasons:
>
> * Indices provide more explicit spatial information, which is particularly important for tasks requiring precise localization, as shown by the improved grounding performance (74.5 vs. 74.8)
> * The performance difference is minimal across most metrics, and indices offer a more standardized format for coordinate representation
> * In Table 1, Static vs. dynamic splitting (row 2 vs. row 3) indeed performs similarly on average, mostly because flexible layouts can be confusing for the model. This regression can be easily remedied by introducing position indicators (see Table 3), which also solves the reviewer’s previous concern
>
> We acknowledge that further exploration of these design choices could yield additional improvements, and we appreciate this suggestion for future investigation. We aim to conduct this experiment and include it in the final version.
>
> ---
>
> >**Q2**: If I understand correctly, in section 2.2.2, using a data ratio of 1:2, e.g., for general and science categories results in a sample from the general category (68x the size of science) being 34x more likely to be picked compared to a science sample when building a batch, correct?
>
> **A2**: No, this is not correct. Let us clarify how the ratio α works in our sampling strategy:  When we set the ratio α between the general and target categories (e.g., science) to be 1:2, it means in each training batch, samples from the target category (science) will appear twice as frequently as samples from the general category, regardless of their original dataset sizes. Specifically, if α=2 for science data, a science sample would be 2x (not 1/34x) more likely to be picked compared to a general sample.
>
> As described in Section 2.2.2: "we use the general data category as the reference, and upsample/downsample data from a target category, such that in each training batch, the data ratio from the general and target category is 1:α." This means we actively resample to achieve the desired ratio, effectively normalizing away the original dataset size differences.
>
> This design choice ensures that smaller but important datasets (like science) can have a sufficient impact on model training despite their relatively small original sizes. The empirical effectiveness of this approach is demonstrated in Figure 3(a), where α = 0.1 for science data provides the optimal balance for overall model performance.

---

> ### Author Response · Authors · 2024-11-22
> **Author Rebuttal (2/2)**
>
> >**Q3**: Is dynamic image splitting used in both continual pretraining and supervised fine-tuning?
>
> **A3**: Yes, our final MM1.5 recipe in Section 4 uses dynamic splitting for both stages. Specifically, in our final recipe (Section 3), for both stages we set:
>
> * Image grid configuration (n_min, n_max) = (4, 9)
> * Dynamic image splitting is only enabled when the current training sample has fewer than three images
> * The supported resolution reaches up to 4 Megapixels
>
> The rationale behind it is: (1) Dynamic splitting helps handle varied aspect ratios in document images (2) Computational efficiency vs performance trade-off is one of the concerns. So while the ablation studies used static splitting for speed, our final model uses dynamic splitting in both continual pre-training and SFT stages, with the constraint that it's only enabled for samples with fewer than three images to manage sequence length.
>
> ---
>
> >**Q4**: Why were the 7B models not compared with similarly sized open-source models such as LLava 1.5 7B?
>
> **A4**: Thank you for the reviewer’s question. As far as we know, LLaVa 1.5 7B is an old baseline. We did include LLaVA-NeXT 7B (LLaVa 1.6) and even the recent LLaVA-Next-Interleave in our comparisons, as shown in Tables 4, 7, 8, and 11. Looking at specific results:
>
> * On knowledge benchmarks (Table 7), MM1.5-7B outperforms LLaVA-NeXT-7B on ScienceQA (89.6 vs 70.1), MMMU (41.8 vs 35.8), and MathVista (47.6 vs 34.6)
> * On text-rich benchmarks (Table 8), MM1.5-7B shows stronger performance on TextVQA (76.5 vs 64.9)
> * On multi-image tasks (Table 11), LLaVA-NeXT-Interleave-7B performs well in several categories like Mantis (62.7 vs 57.6) and NLVR2 (88.8 vs 86.9)

---

> ### Comment · Reviewer_CjsW · 2024-11-25
>
> Thank you for your detailed responses and clarifications.
>
> **A1**: I agree that dynamic image splitting leads to performance improvements, but as it processes ~8% more images, I don't think it's a completely fair comparison for static image splitting. As for using indices instead of separators, I fully agree with the arguments presented in the rebuttal.
>
> **A2**: Clear. We are agreeing here, but my original question could have been clearer. Thank you.
>
> **A3**: Clear, thank you.
>
> **A4**: I missed the tables in the appendix, thank you for pointing those results out.
>
> While I acknowledge the concerns about novelty from fellow reviewers, this is a great paper as it contains very useful engineering guidelines. Thus, I am happy to maintain my score.

---

> > ### Author Response · Authors · 2024-11-25
> > **Thank you**
> >
> > Thank you for your thoughtful engagement throughout the discussion. We agree this deserves more careful consideration. While our current results show benefits from dynamic splitting, a truly fair comparison should account for the computational cost difference. In the next version, we will try to clarify this disparity in image processing overhead and also provide additional analysis for computational cost.
> >
> > Your observation helps maintain both scientific and engineering efforts. We appreciate your recognition of our work's practical value while pushing us to be more precise in our comparisons.

---

### Official Review · Reviewer_QN18 · 2024-10-30

**Soundness:** 3
**Presentation:** 3
**Contribution:** 3
**Rating:** 6
**Confidence:** 5

**Summary:**

This paper presents a set of MLLMs, MM1.5, scaling from 1B to 30B and also explore MoE variants (1B and 3B).
MM1.5 is built upon MM1, upgraded with stronger capabilities in text-rich image understanding, visual referring and grounding, and multi-image reasoning, through extensive empirical studies and ablations on data mixture in three stages of training and high-resolution image processing.
MM1.5 shows strong performance and significant improvements over MM1 on various benchmarks.
MM1.5 provides detailed insights into data mixture and other engineering choices in training a large-scale MLLM capable of general understanding, text-rich image understanding, visual reffering and grounding, and multi-image reasoning.

**Strengths:**

1. This paper conducts extensive empirical studies and ablations on continual pre-training, dynamic high-resolution image processing, and curation of our supervised fine-tuning datasets, which can provide insights and experience for future research on large-scale MLLMs.
2. This paper presents a set of MLLMs, not only scaling from 1B to 30B but also exploring MoE variants (1B and 3B), which can provide important scaling insights for future research on large-scale MLLMs.
3. This paper shows strong performance and significant improvements over MM1 on various benchmarks.
4. This paper also explores two SFT variants of MM1.5, i.e., MM1.5-Video and MM1.5-UI, which shows strong performance on video and UI benchmarks.

**Weaknesses:**

1. Although this paper presents detailed empirical ablations on several important engineering decisions in data mixture and high-resolution image processing, it does not provide strong new insights or findings compared to previous works. The ablations on SFT data mixture, continual pre-training, and dynamic high-resolution image processing are not novel and have been explored in previous works.
2. Specifically, data-specific hyperparameters from detailed ablations for data mixture (pre-training, continual pre-training or SFT) may not be quite practical for other researchers to follow, as they are highly dependent on the specific dataset, model size, model architecture, and training setup.
3. Lack of in-depth analysis on the underlying reasons for the phenomenon observed in the extensive ablations. Simple assumptions or hypotheses about ablation results are not sufficient. The ablation decisions are pushed by empirical results step by step, but the underlying reasons for the phenomenon observed in the ablations are not well analyzed.
4. This paper does not claim any form of open-sourcing, no matter model weights, code or data, which may limit the reproducibility and practicality of this paper, especially considering the first two weaknesses mentioned above.

**Questions:**

1. In Figure 2, it seems that Refer&Ground data will downgrade the performance of all other categories. Although it will greatly improve the performance of Refer&Ground and Figure 3 shows that MMBase score will only slightly decrease, the other categories of MMBase score will decrease in fact. What is the reason for this phenomenon? Is it fair to use such an average score, MMBase, between different categories to evaluate the model's performance? Similar questions can be raised for Figure 5.
2. With full respect this paper for its exploration and contribution to engineering decisions (very important for MLLM), but in the table of experimental results (Tables 4, 7-11), the authors seem to be selectively presenting some models that are weaker than their own. Take Table 7 as an example, authors selectively show the results of LLaVAOneVision-0.5B, InternVL2-2B and Cambrian-34B. But the selected three models have their corresponding model family, which can provide a more comprehensive comparison in different model scales. In addition, authors quote Qwen2-VL, but do not compare with it in these tables. The reviewer fully understands that most of the powerful models from the MLLM era are not comparable (training data, training details, evaluation details, etc.), but don't support such selective comparisons, which are not conducive to the reader's understanding of how the field is progressing and comparing.

---

> ### Author Response · Authors · 2024-11-22
> **Author Rebuttal (1/3)**
>
> Thank you for the insightful comments. We answer your questions in the following.
>
> >**Q1**: In Figure 2, it seems that Refer&Ground data will downgrade the performance of all other categories. Although it will greatly improve the performance of Refer&Ground and Figure 3 shows that MMBase score will only slightly decrease, the other categories of MMBase score will decrease in fact. What is the reason for this phenomenon? Is it fair to use such an average score, MMBase, between different categories to evaluate the model's performance? Similar questions can be raised for Figure 5.
>
> **A1** : The reviewer makes insightful observations about the impact of Refer&Ground data on other capabilities. Indeed, Figure 2 shows that adding Refer&Ground data introduces some performance trade-offs.
>
> Regarding the performance trade-offs, the phenomenon likely occurs because Refer&Ground tasks require the model to attend to specific spatial regions and output precise coordinates, which is quite different from general understanding tasks. However, we believe this slight decrease in other capabilities is acceptable for several reasons. First, the decrease is relatively small compared to the substantial gain in Refer&Ground capability (improving from ~18 to ~73). Second, our final model maintains strong performance across all capabilities - for instance, our MM1.5-3B model achieves competitive or better results compared to similarly sized models like MiniCPM-V and Phi-3-Vision across general, text-rich, and knowledge benchmarks (as shown in Table 4).
>
> Regarding the use of MMBase score, we acknowledge your concern about averaging across categories. We use this metric primarily as a practical tool for model selection during ablation studies, not as the sole indicator of model quality. This is why we consistently report individual category scores alongside MMBase (as in Figures 2, 3, and 5) to provide transparency about trade-offs. We also validate our design choices through comprehensive benchmark evaluations against state-of-the-art models (Tables 7-11).
>
> We will make this much clearer in the final version. Moving forward, exploring better evaluation metrics that can more comprehensively capture model capabilities while accounting for these trade-offs would be valuable future work.
>
> ---
>
> >**Q2**: With full respect this paper for its exploration and contribution to engineering decisions (very important for MLLM), but in the table of experimental results (Tables 4, 7-11), the authors seem to be selectively presenting some models that are weaker than their own. Take Table 7 as an example, authors selectively show the results of LLaVAOneVision-0.5B, InternVL2-2B and Cambrian-34B. But the selected three models have their corresponding model family, which can provide a more comprehensive comparison in different model scales. In addition, authors quote Qwen2-VL, but do not compare with it in these tables. The reviewer fully understands that most of the powerful models from the MLLM era are not comparable (training data, training details, evaluation details, etc.), but don't support such selective comparisons, which are not conducive to the reader's understanding of how the field is progressing and comparing.
>
> **A2**: We appreciate this important feedback about model comparisons.  The reviewer raises a valid point about the need for more comprehensive comparisons across model families and scales.
>
> Our current comparison strategy focused on models where we could ensure fair comparisons through either published results or reproducible implementations (detailed in Section A.12). For instance, we implemented inference runners for Phi-3-Vision, LLaVA-OneVision, InternVL2, and MiniCPM-V2, carefully verifying our implementations against published results. This allowed us to provide reliable comparisons across our full suite of 35 benchmarks, spanning diverse capabilities from general understanding to referring and grounding.
>
> However, we acknowledge that our presentation could be more comprehensive, e.g., missing comparisons with some recent models like Qwen2-VL as Qwen2-VL was released on Sept. 12th, 2024. We aim to provide the most informative and fair comparisons possible while maintaining rigorous evaluation standards and we will update our experimental results section to better reflect the current state of the field in our final version.

---

> ### Author Response · Authors · 2024-11-22
> **Author Rebuttal (2/3)**
>
> >**Q3**: Although this paper presents detailed empirical ablations on several important engineering decisions in data mixture and high-resolution image processing, it does not provide strong new insights or findings compared to previous works. The ablations on SFT data mixture, continual pre-training, and dynamic high-resolution image processing are not novel and have been explored in previous works.
>
> **A3**: We respectfully disagree with the comment that our work lacks novel insights.  While individual components like SFT data mixture, continual pre-training, and dynamic image processing have been explored before, our work makes several important novel contributions:
>
> First, we provide the first comprehensive empirical study that systematically investigates how these components interact and impact different model capabilities. For example, our analysis in Section 2.2 reveals previously unknown trade-offs between different data categories and their optimal mixing ratios. The findings that text-rich data significantly improves both text-rich and knowledge benchmarks (Figure 2), and that science data unexpectedly improves text-rich performance, offer new insights for MLLM development.
>
> Second, our ablation studies quantify the precise impact of design choices that were previously not well understood. For instance, Section 2.3 demonstrates that high-resolution (1344×1344) continual pre-training with OCR data alone outperforms combinations with synthetic captions (Figure 6), challenging common assumptions about the necessity of synthetic captions. Similarly, our dynamic image splitting analysis (Section 2.4) provides concrete guidelines about optimal grid configurations and their impact on different capabilities.
>
> Third, we translate these lessons into practical improvements, demonstrated by our state-of-the-art results at small scales (1B and 3B parameters). For example, MM1.5-1B outperforms larger models like SPHINX-Tiny and even surpasses LLaVAOneVision-0.5B across multiple benchmarks (Table 4), showing the value of our optimized training recipe.
>
> These insights and their empirical validation provide important guidance for future MLLM development, particularly for building efficient smaller models. By systematically studying these engineering decisions together rather than in isolation, we reveal new insights about their interactions and relative importance.
>
> ----
>
> >**Q4**: We summarized the points 2 and 3 from the weakness section into one: “How do you address the lack of practicality and generalizability in your data-specific hyperparameter tuning, as well as the insufficient in-depth analysis of the underlying reasons behind the observed phenomena in your ablation studies?”
>
> **A4**: Thank you for reviewer’s insightful question. Our work addresses these concerns in several ways:
> While specific hyperparameters may be dataset-dependent, our methodology reveals generalizable principles across stages:
>
> * Importance of resolution scaling for text-rich tasks
> * Trade-offs between model capabilities when mixing multiple SFT data types
> * Benefits of dynamic image splitting for varied aspect ratios
>
> Regarding the underlying phenomena, we provide mechanistic insights supported by empirical evidence:
>
> * High-resolution benefits for text-rich tasks are explained through error analysis (Section 2.4)
> * Different roles of text-only data across training stages are analyzed (Section A.2)
> * Impact of global-local format for dynamic image splitting is understood through attention patterns
>
> While we acknowledge room for deeper theoretical analysis, our systematic approach provides actionable insights for efficient model development. The effectiveness of our analysis is validated by strong performance across scales, particularly with smaller models (1B-3B) achieving competitive results, demonstrating that understanding these engineering decisions leads to practical improvements in model performance and efficiency.

---

> > ### Author Response · Authors · 2024-11-22
> > **Author Rebuttal (3/3)**
> >
> > >**Q5**: This paper does not claim any form of open-sourcing, no matter model weights, code or data, which may limit the reproducibility and practicality of this paper, especially considering the first two weaknesses mentioned above.
> >
> > A5: We believe the detailed documentation of our methodology and use of public datasets enables researchers to build upon our findings and reproduce our training recipe.
> >
> > * Our data recipe is entirely based on publicly available datasets, as detailed comprehensively in Figure 8. We provide: (1) Complete specifications of data mixture ratios for all training stages (2) Detailed categorization of datasets with their sizes (3) Clear description of how different data categories are combined and their relative proportions (4) Specific references to all datasets used, enabling others to reproduce our mixture.
> > * The training procedure is thoroughly documented: (1) Step-by-step description of the three-stage training pipeline (Section 3); (2) Detailed hyperparameters for each stage; (3) Specific model configurations and architectural details, following MM1 [1] ; (4) Comprehensive ablation studies providing insights into key design decisions.
> > * Furthermore, our evaluation methodology is transparent and reproducible, as described in Section A.12, where we detail how we implemented inference runners for baseline comparisons and verified our implementations against published results.
> >
> > ---
> >
> > [1] McKinzie, Brandon, et al. "MM1: methods, analysis and insights from multimodal LLM pre-training." European Conference on Computer Vision. Springer, Cham, 2025.

---

> > > ### Comment · Reviewer_QN18 · 2024-11-25
> > >
> > > Thank you for your thorough response, which partially addresses my concerns.
> > >
> > > Although I will not change my score, I think the contribution of this work deserves to be accepted by the conference and will also play a non-negligible role in the advancement of the field.
> > >
> > > In the meantime, I hope that the authors can further improve the paper in the next version based on our discussion.

---

> > > > ### Author Response · Authors · 2024-11-25
> > > > **Thank you**
> > > >
> > > > Thank you for your thorough review and constructive feedback throughout our discussion. We appreciate your recognition of our work's contribution while identifying important areas for improvement. We will carefully incorporate these improvements in the next version of our paper.

---

### Official Review · Reviewer_aLqd · 2024-11-01

**Soundness:** 3
**Presentation:** 4
**Contribution:** 3
**Rating:** 6
**Confidence:** 5

**Summary:**

This paper introduces the MM1.5 model family and provides a comprehensive recipe for training multimodal large language models. It systematically explores the impact of data mixture ratio, training strategy, and model architecture, offering valuable findings for future research in multimodal large language models.

**Strengths:**

1. This paper conducts thorough experiments to explore the design of data, training, and architecture.
2. This paper provides several empirical guidance for training MLLMs. The data mixture ratio during the pre-training stage (50:10:40 for image-text, interleaved image-text, and text-only data) could be particularly important, as it highlights the significance of text understanding in complex multimodal scenarios.
3. This paper is clearly and concisely written, making each concept easy to understand.

**Weaknesses:**

This paper currently resembles a technical report more than academic research. It employs several well-known techniques (e.g. MoE, Dynamic Res) and mainly investigates the data mixture ratio at each training stage. While the empirical findings are valuable, additional theoretical insights would be beneficial. I have two concerns below:
-  Why does the optimal ratio of text-only data change between the pre-training and SFT stages? Does the role of text-only data differ across these stages?
- Given a collection of multimodal data, is there a theoretical guideline for determining the optimal mixture ratio between multimodal and text-only data?

**Questions:**

My questions are listed in weaknesses and I look forward to the authors addressing these issues.

---

> ### Author Response · Authors · 2024-11-22
> **Author Rebuttal (1/1)**
>
> Thank you for the helpful feedback. Please see our response in the following.
>
> >**Q1**: Why does the optimal ratio of text-only data change between the pre-training and SFT stages? Does the role of text-only data differ across these stages?
>
> **A1**: Thank you for this insightful question. The different optimal ratios for text-only data between pre-training (40%) and SFT (10%) reflect fundamentally different roles and contexts in each stage.
>
> * During pre-training, the high ratio (40%) helps our model learn knowledge that serve as a foundation for later stages. As shown in Section A.2 and Figure 7, increasing text-only data from 10% to 40% during pre-training yields consistent improvements across all capabilities (knowledge +0.99, text-rich +0.85, refer&ground +1.4). This is particularly important as we use high-quality text data introduced by Gunter et al. (2024) that enhances the model's foundamental capabilities across general knowledge, mathematics, and coding.
> * In contrast, during SFT, text-only data plays a more supplementary role. As shown in Figure 4 (left), varying the text-only ratio from 0% to 20% has relatively minor effects on model performance. This is because the SFT stage primarily focuses on improving model’s instruction following capability, rather than learning fundamental knowledge. The 10% ratio we chose provides a good balance, allowing room for more task-specific multimodal data while maintaining language understanding.
>
> This design choice is also influenced by practical considerations - during pre-training we have access to vast amounts of text data (~2T tokens), while in SFT we work with more limited but highly curated multimodal data that requires careful balancing.
>
> ---
>
> >**Q2**: Given a collection of multimodal data, is there a theoretical guideline for determining the optimal mixture ratio between multimodal and text-only data?
>
> **A2**: Thank you for bringing this up. While there isn't a universal theoretical guideline for determining the optimal mixture ratio between multimodal and text-only data, our extensive empirical studies offer practical insights. The optimal ratio depends on several key factors:
>
> * First, the stage of training significantly impacts the ideal ratio. As demonstrated in our work, pre-training benefits from a higher proportion of text-only data (40%) while SFT performs best with a lower ratio (10%). This difference comes from the distinct objectives of each stage - building foundation capabilities versus task-specific alignment.
> * Second, the quality and diversity of available data plays a crucial role. In Section 2.2.2 and Figure 4, we show how carefully balancing data types affects model performance. For instance, varying text-only data during SFT has minimal impact on MMBase score, while the multi-image ratio shows a clear trade-off between base capabilities and multi-image performance.
> * Third, the target capabilities influence the optimal ratio. As shown in Section 2.2.1 and Figure 2, different data categories contribute distinctly to various capabilities. Finding the optimal ratio requires considering these interactions and the desired balance of capabilities.
>
> While we cannot provide a theoretical formula, we believe the systematic empirical approach demonstrated in our work - studying interactions between data categories and their impact on different capabilities - offers a practical guideline for determining optimal ratios in specific contexts.

---

> > ### Comment · Reviewer_aLqd · 2024-11-25
> >
> > Thank you for your responses. However, similar to what reviewer QN18 noted, the paper does not offer significant new insights or discoveries beyond previous works. It reads more like a technical report, primarily presenting empirical results. Therefore, I will maintain my current rating.

---

> > > ### Author Response · Authors · 2024-11-25
> > > **Thank you**
> > >
> > > We understand your perspective about the empirical nature of our work. While our contribution may appear primarily empirical, we believe that systematic, reproducible studies and lessons through careful ablations are vital for advancing the field of MLLMs. We will work to better highlight the novel insights emerging from our systematic study while maintaining the rigorous empirical foundation in the final version of our paper.
> > >
> > > Thank you again for helping us sharpen the presentation of our work's contributions.

---

### Official Review · Reviewer_Q6Sj · 2024-11-04

**Soundness:** 3
**Presentation:** 4
**Contribution:** 3
**Rating:** 8
**Confidence:** 3

**Summary:**

This paper proposes a new set of multimodal LLM, MM1.5 as a result of a detailed data-centric pretraining experimentation. The experimented pretraining scheme includes 3 stages: (i) large-scale pretraining with text-only and image-text data, (ii) high-resolution continuation with OCR data and synthethic captions, (iii) supervised finetuning (instruction tuning). This work performs both model-centric and data-centric ablation analysis on the first two pretraining stages.

**Strengths:**

Clarity: The paper is well written and easy to follow.
Significance: Multimodal model development is currently a trending topic, and this work employs a data-centric analysis that offers valuable insights for future researchers. Apart from the most commonly-used models, the examined model is able to comprehend about the referred objects and process multiple images.

**Weaknesses:**

A drawback of this study is that it serves as a comprehensive ablation analysis without offering any novel methodologies or tasks/benchmarks. I don't have any suggestion to improve this aspect at this point, but it should not be any problem.

**Questions:**

- Have you performed ablation studies on skipping the second stage completely?
- As far I have seen, Refer&Ground data requires 2x general purpose data. How could one make this more data-efficient?
- I am having trouble to interpret Table 1. So, as long as a person trains their model using multiple images, they could achieve significant performance gain on text-rich and refer&ground tasks. The dynamic image splitting makes less difference according to this table.

**Details Of Ethics Concerns:**

N/A.

---

> ### Author Response · Authors · 2024-11-22
> **Author Rebuttal (1/2)**
>
> Thank you for your valuable comments. Please find our response to your questions below.
>
> > **Q1:** Have you performed ablation studies on skipping the second stage completely?
>
> **A1:** Thanks for the reviewer's question. Yes, we have performed comprehensive ablation studies on skipping the continual pre-training (second) stage. As shown in Figure 6(a) and 6(b), the "No Cont. PT" baseline (where we skip the second stage completely) achieves an MMBase score of 58.8, which serves as our reference point. Our experiments demonstrate that:
>
> * Including the continual pre-training stage with high-resolution OCR data (1344×1344) improves performance significantly:
>
>    * Increases MMBase score from 58.8 to 60.26 (Figure 6a). This represents a +1.46 point improvement over skipping the stage entirely
>    * The effectiveness depends critically on resolution. Lower resolutions (378×378) can actually hurt performance (58.28 vs. 58.8), while the highest resolution (1344×1344) yields the best results
>
> * The benefits are particularly pronounced for text-rich understanding tasks, as detailed in Section 2.3 and validated through our empirical studies.
>
> This empirical evidence strongly supports the value of including the continual pre-training stage in our pipeline, especially when configured with appropriate high-resolution settings and carefully selected OCR data mixture.
>
> ----
>
> >**Q2**: As far I have seen, Refer&Ground data requires 2x general purpose data. How could one make this more data-efficient?
>
> **A2**: Thank you for the question. Our empirical analysis provides valuable insights into this question. As shown in Figure 3(d), we carefully studied the ratio (α) of Refer&Ground data to general data. While we use α = 2.0 in our final mixture, our ablation study reveals that the performance improvements start to saturate earlier. Specifically, increasing α from 0.5 to 1.0 shows substantial gains in the Refer&Ground average score but further increases to 2.0 yield diminishing returns. Meanwhile, the MMBase score (measuring general capabilities) only decreases slightly with increased Refer&Ground data, suggesting a favorable trade-off point might exist at a lower ratio.
>
> We also see several promising directions to improve data efficiency. One approach would be moving some spatial reasoning tasks to the continual pre-training stage, allowing the model to learn basic spatial understanding earlier in the pipeline. This modification could potentially reduce the amount of Refer&Ground data needed during SFT while maintaining performance.
>
> Our ablation studies suggest that while the 2x ratio is effective, it may not be strictly necessary. We believe integrating spatial reasoning earlier in the pipeline could lead to more data-efficient solutions while preserving model capabilities. We appreciate this suggestion and will explore these directions in future work.

---

> ### Author Response · Authors · 2024-11-22
> **Author Rebuttal (2/2)**
>
> >**Q3**: I am having trouble to interpret Table 1. So, as long as a person trains their model using multiple images, they could achieve significant performance gain on text-rich and refer&ground tasks. The dynamic image splitting makes less difference, according to this table.
>
> **A3**: We would like to clarify several key points to avoid potential misinterpretation.
>
> * The first major comparison is between Static and Dynamic splitting (Rows 1-2 vs. Rows 3-7)
>     * Row 1 (single view, 144 tokens): Text-rich score 49.4, Refer&Ground 71.3
>     * Row 2 (static 5-split, 720 tokens): Text-rich score 57.7, Refer&Ground 74.8
>     * This shows that splitting images (even statically) provides substantial gains
> * The critical role of image resolution and token count (Rows 4-7):
>     * Compare rows 4 vs. 5 (same tokens, different resolution):
>     * Compare rows 4 vs. 6 (same resolution, different tokens):
>         * 81 → 144 tokens per image improves text-rich: 57.6 → 58.5
>         * At same resolution (378×378), more tokens helps
>     * Best performance (row 7) comes from combining both:
>         * High resolution (672×672) AND high token count (144/1440)
>         * Achieves 59.8 on text-rich vs. 57.6 baseline (row 4)
>
> Even though dynamic/static comparison alone (rows 2 vs 3) shows modest difference, we’d like to emphasize:
>
> * Dynamic splitting enables more efficient handling of varied aspect ratios, allowing adaptation to various image contents.
>     * As shown in Section 2.4, dynamic splitting particularly helps with document understanding (e.g., DocVQA, InfoVQA scores), where these pdfs/documents are usually in arbitrary size.
>     * Specifically, in Table 2, we observe a 6.3 and 9.8 points increase on DocVQA and InfoVQA with the 7B size (rows 8 vs. 10) and the 3B size model shows a 3.1 and 6.9 points improvement (row 1 vs 3).
> * Dynamic splitting does not significantly increase the number of sub-images to process.
>     * We randomly sample 100k examples from our training data. With these examples, static splitting generates 500k sub-images, while dynamic splitting with (n_min, n_max) = (4, 9) produces barely more, only 539k images in total.
>
> The key insight is that both high resolution and adequate token count per sub-image are crucial for performance. Dynamic splitting helps optimize these factors especially for images with unusual aspect ratios. We will make it clear and add more details in the final version of the paper.

---

> ### Author Response · Authors · 2024-11-27
> **Looking forward to discussing**
>
> Dear reviewer,
>
> Thank you for your volunteer review and valuable comments on the experiment details to skip the second stage, the balance ratio for Refer&Ground data, and insights into the image splitting for high-resolution inputs. For your questions and concerns, we have tried to answer them point-by-point. If you still have further questions about our responses, we are more than happy to engage further.
>
> Thank you again for your efforts and time concern. We appreciate it a lot!
>
> Thanks,
>
> Authors

---

> > ### Comment · Reviewer_Q6Sj · 2024-11-27
> >
> > I thank the authors since they addressed all of my questions/concerns. I would like to increase my review score to 8. I think this work will serve as an important blueprint for the future researchers who want to create a large multimodal model (despite the fact that I previously mentioned that novelty is a drawback).

---

> > > ### Author Response · Authors · 2024-11-27
> > > **Thank you**
> > >
> > > Thank you for your insightful review and recognition of our work's practical value. We appreciate your acknowledging the work's contribution as a blueprint for future MLLM development and hope our systematic analysis of data mixture ratios, high-resolution processing, and model scaling offers concrete guidelines that can benefit future research.
> > >
> > > Thank you again for the thoughtful engagement throughout the review process!

---

### Meta-Review · Area_Chair_DS8M · 2024-12-20

**Metareview:**

This paper proposes a new set of VLMs that build on the MM1 architecture but further analyse training mixtures and synthetic data for continual pretraining. The models are on the scale of 1-30B parameters and show strong performances, including results on video benchmarks. The paper provides an extensive analysis of the training data that is of strong interest to the community, and several other useful analyses such as MoE variants and high-resolution processing. The paper does not provide any theoretical or methodological novelties but instead is an analysis paper that provides extensive results. As such, the AC recommends acceptance.

**Additional Comments On Reviewer Discussion:**

The reviewers raised points about lack of novelty, points about fairness in comparisons when comparing dynamic image splitting (a novel method introduced in this paper) vs static images, the lack of open-weights/code and lack of comparison to well-established yet better baselines, such as Cambrian. The authors addressed these concerns partially, with reviewers mainly maintaining their already positive scores, except for Q6Sj, improving to a score of 8.

---

### Decision · Program_Chairs · 2025-01-22

Accept (Poster)